# *Plasmodium falciparum* impairs Ang-1 secretion by pericytes in a 3D brain microvessel model

Rory K M Long [ID][1,2], François Korbmacher[1], Paolo Ronchi [ID][3], Hannah Fleckenstein[1,4], Martin Schorb[3], Waleed Mirza[1], Mireia Mallorquí[1], Ruth Aguilar[5], Gemma Moncunill [ID][5], Yannick Schwab[3,4] & Maria Bernabeu [ID][1,✉]

## Abstract

Disruption of the vascular protective angiopoietin–Tie axis is common in cerebral malaria (CM) patients, who display elevated angiopoietin-2 (Ang-2) and reduced angiopoietin-1 (Ang-1) blood concentrations. The role of pericytes in CM pathogenesis remains unexplored, despite being a major source of brain Ang-1 secretion and evidence of pericyte damage observed in CM postmortem samples. Here, we engineered a human 3D microfluidics-based brain microvessel model containing the minimal cellular components to replicate the angiopoietin–Tie axis, human primary brain microvascular endothelial cells, and pericytes. This model replicated pericyte vessel coverage and ultrastructural interactions present in the brain microvasculature. When exposed to *P. falciparum*-iRBC egress products, 3D brain microvessels presented decreased Ang-1 secretion, increased vascular permeability, and minor ultrastructural changes in pericyte morphology. Notably, *P. falciparum*-mediated barrier disruption was partially reversed after pre-treatment with recombinant Ang-1 and the Tie-2 activator, AKB-9778. Our approach suggests a novel mechanistic role of pericytes in CM pathogenesis and highlights the potential of therapeutics that target the angiopoietin–Tie axis to rapidly counteract vascular dysfunction caused by *P. falciparum*.

**Keywords** Cerebral Malaria; 3D Brain Microvascular Models; Pericytes; Angiopoietin–Tie Axis; AKB-9778
**Subject Categories** Microbiology, Virology & Host Pathogen Interaction; Neuroscience; Vascular Biology & Angiogenesis

## Introduction

Cerebral malaria (CM) is a severe neurological complication of *Plasmodium falciparum* infection, clinically characterized by coma and fatality rates averaging 15–20% (Dondorp et al, 2010). Furthermore, approximately half of CM survivors endure lifelong neurological sequelae such as hemiplegia, ataxia, epilepsy, or speech disorders (Ray et al, 2025). The majority of pediatric CM-related deaths present severe brain swelling due to vasogenic edema, likely as a consequence of blood–brain barrier dysfunction (Mohanty et al, 2017). A hallmark of CM is the sequestration of *P. falciparum*-infected red blood cells (iRBC) in the brain microvasculature (Dorovini-Zis et al, 2011). While the molecular players responsible for *P. falciparum*-iRBC ligand–receptor interactions with cerebral endothelial cells have been extensively studied (Kessler et al, 2017; Storm et al, 2019; Sahu et al, 2022), the consequences of iRBC sequestration in the brain microvasculature require further investigation. Nevertheless, evidence suggests that increases in vascular permeability during CM are a result of a multifaceted process including the blockade of barrier-supportive endothelial receptors by iRBC, the release of toxic *P. falciparum* products from egressed iRBC, and a dysregulated host innate immune response (Bernabeu and Smith, 2017; Wassmer et al, 2024).

An important pathway commonly dysregulated in CM patients is the angiopoietin–Tie axis, crucial to maintaining endothelial pro-barrier, anti-inflammatory, and anti-apoptotic functions (Augustin et al, 2009). Binding of the ligand angiopoietin-1 (Ang-1) to the endothelial receptor Tie-2 promotes vascular quiescence and the stabilization of endothelial tight junction proteins, such as claudin-5, occludin, and zona occludens-1 (ZO-1), necessary for maintaining the barrier function of the brain microvasculature (Fukuhara et al, 2008; Saharinen et al, 2008; Siddiqui et al, 2015). Conversely, angiopoietin-2 (Ang-2), which is rapidly released from small endothelial storage granules called Weibel-Palade bodies, generally acts as a Tie-2 antagonist leading to endothelial activation and vascular leakage (Maisonpierre et al, 1997). Given the opposing roles of these two molecules in vascular function, decreased levels of Ang-1 and increased levels of Ang-2 have been largely associated with cerebral vascular pathogenesis in a multitude of diseases (Gurnik et al, 2016; Golledge et al, 2014). Similarly, a decrease in Ang-1 and an increase in Ang-2 and the Ang-2:Ang-1 ratio have been well-documented in fatal cases of both pediatric and adult CM (Yeo et al, 2008; Lovegrove et al, 2009; Conroy et al, 2009, 2012; Jain et al, 2011). Despite its importance, the mechanisms leading to angiopoietin–Tie axis disruption in CM are poorly understood. Human pro-inflammatory cytokines and pro-coagulation proteins,

[1]European Molecular Biology Laboratory (EMBL) Barcelona, Barcelona, Spain. [2]Heidelberg University, Faculty of Biosciences, Heidelberg, Germany. [3]Electron Microscopy Core Facility, European Molecular Biology Laboratory (EMBL), Heidelberg, Germany. [4]European Molecular Biology Laboratory (EMBL), Cell Biology and Biophysics Unit, Heidelberg, Germany. [5]Barcelona Institute for Global Health (ISGlobal), Hospital Clínic-Universitat de Barcelona, Barcelona, Spain. ✉E-mail: maria.bernabeu@embl.es

such as TNF-α and thrombin, have been proposed to be responsible for the increase of Ang-2 secretion by endothelial cells (Gomes et al, 2023; Fiedler et al, 2004), but the causative agent responsible for decreased secretion of Ang-1 in CM patients remains unknown.

Vascular mural cells, including smooth muscle cells and pericytes, are the main cellular sources for Ang-1 secretion in the brain microvasculature (Augustin et al, 2009). Beyond promoting endothelial barrier function through Ang-1 secretion, pericytes support the microvasculature and promote quiescence through additional roles. Structurally, pericytes wrap around cerebral microvessels and interact with the endothelium through reciprocal ultrastructural membrane protrusions, known as peg-and-socket junctions (Ornelas et al, 2021), and secrete extracellular matrix components of the basal lamina that strengthen microvascular architecture. Functionally, pericytes regulate brain blood flow by controlling vessel diameter through constriction (Daneman and Prat, 2015). Cerebral pericytes have been implicated in various vascular disorders of the central nervous system. Loss of pericytes from the cerebral vasculature during ischemic stroke and Alzheimer's disease is associated with brain microvascular disruption, extravasation of blood components into the brain parenchyma, and neuronal loss (Fernández-Klett et al, 2013; Sengillo et al, 2013). Similarly, histopathological studies in retinal samples of fatal pediatric CM cases have revealed pericyte damage in regions of iRBC sequestration, suggesting a similar role of these cell types in severe malaria infections (Barrera et al, 2018). Yet, whether cerebral pericyte dysfunction promotes CM pathogenesis and the dysregulation of the angiopoietin–Tie axis remains largely unknown. Bioengineered vascular models are emerging as powerful tools to study disease mechanisms in vitro. They have been successfully used to recapitulate important vascular pathways, including the role of pericytes and the angiopoietin–Tie axis in the establishment of vascular networks (Haase et al, 2019), and more recently, malaria pathogenesis, by modeling ligand–receptor interaction of iRBC in microvessels (Bernabeu et al, 2019) or parasite-induced endothelial stress and inflammatory response (Howard et al, 2023; Piatti et al, 2025).

Here, we have developed a 3D brain microvascular model composed of primary human brain endothelial cells and pericytes that recapitulates the minimal functional cellular unit of the angiopoietin–Tie axis found in the in vivo human brain. We have used this model to understand the interplay between iRBC, brain endothelial cells and pericytes that contributes to the molecular basis of angiopoietin–Tie dysregulation in CM. Our results indicate that products released during iRBC egress play a direct role in angiopoietin–Tie dysregulation by decreasing Ang-1 secretion from pericytes, highlighting this cell type as a new player in CM pathogenesis. Furthermore, we showed that parasite-induced increase in endothelial permeability can be therapeutically targeted by the use of the angiopoietin–Tie axis inducers, such as the Tie-2 activator AKB-9778.

## Results

### In vitro 3D brain microvessel model recapitulates in vivo cerebral endothelial–pericyte interactions

Primary human brain microvascular endothelial cells (HBMEC) and human brain vascular pericytes (HBVP) were commercially purchased and the expression of cell type-specific markers was validated, including junctional proteins and transporters: vascular endothelial (VE)-cadherin, β-catenin, zonula occludens-1 (ZO-1), claudin-5, and glucose transporter-1 (GLUT-1), as well as, endothelial-specific markers: CD31 and von Willebrand Factor (vWF) for HBMEC (Fig. EV1A,B). HBVP expressed brain pericyte markers platelet-derived growth factor receptor β (PDGFRβ) and nerve/glial antigen 2 (NG-2) (Fig. EV1C) (He et al, 2016; Attwell et al, 2016). Furthermore, we confirmed exclusive secretion of Ang-1 by HBVP and HBMEC secretion of Ang-2 in a transwell model (Fig. EV1D). To better recapitulate the brain microvascular architecture, we have generated a 3D brain microvessel model that recapitulates the minimal functional cellular unit of the angiopoietin–Tie axis. The device is fabricated in a collagen type I hydrogel containing a microfluidic network with a pre-defined geometry, which is connected to both an inlet and outlet, enabling vessel perfusion (Zheng et al, 2012) (Fig. EV2A). Previous incorporation of pericytes in similar bioengineered 3D models involved cellular addition into the collagen hydrogel itself. However, this resulted in sparse interaction of pericytes with the endothelial microvessels (Zheng et al, 2012). Nevertheless, in vivo cerebral pericytes have been shown to cover approximately one third of the brain microvasculature and present a 3:1 or 1:1 endothelial to pericyte ratio, albeit with slight variations along the brain vascular hierarchical network (Mathiisen et al, 2010; Uemura et al, 2020; Haley and Lawrence, 2017). To recreate this, HBVP were seeded directly into the channels along with HBMEC (Figs. 1A and EV2A). After 3 days in culture, the two cell types reorganized into two different layers with HBMEC forming a network of 100-μm diameter microvessels with HBVP covering the microvessel surface (Figs. 1B and EV2B). To optimize the ratio of endothelial cells to pericytes required to produce perfusable microvessels with high pericyte coverage, we characterized the use of both a 3:1 and 5:1 endothelial cell to pericyte seeding ratio. The 3:1 HBMEC:HBVP ratio resulted in microvessels which displayed significant contraction, resulting in microvessel lumen collapse. In contrast, the 5:1 HBMEC:HBVP ratio formed viable and lumenized microvessels, and therefore, was chosen for subsequent studies (Fig. EV2D). To examine the extent of pericyte coverage in the model, mCherry-expressing HBVPs were utilized in microvessel fabrication. HBVP distributed homogenously around the vessel cross-section, covering approximately a third of the microvessel surface area (29.6% of the bottom surface and 27.3% of the top surface) (Fig. 1B,C). Endothelial VE-cadherin junctional labeling was continuous along the vessel luminal surface, suggesting that the presence of HBVP does not disrupt the endothelial layer (Fig. 1D). Furthermore, HBMEC in the model retained the expression of endothelial adherens and tight junction markers, β-catenin and ZO-1 respectively (Fig. 1E). Consistent with the presence of continuous endothelial junctions, the permeability of 70 kDa FITC dextran was decreased in microvessels containing pericytes (median = $1.23 \times 10^{-6}$ cm/s) compared to HBMEC-only microvessels (median = $1.83 \times 10^{-6}$ cm/s), albeit not significantly (Fig. 1F). HBVP kept the expression of pericyte markers PDGFRβ and NG-2, as well as, the mural cell contractile protein α-smooth muscle actin (αSMA), and appeared as elongated cells with thin processes stretching over multiple endothelial cells (Fig. 1D,E). In addition, HBVP secreted laminin and collagen IV, contributing to the secretion of extracellular matrix components that compose the basal lamina of brain microvasculature in vivo (Fig. 1E) (Oliveira et al, 2023).

Volume electron microscopy has been previously used to characterize in vivo 3D ultrastructural interactions between pericytes and endothelial cells of the brain microvasculature

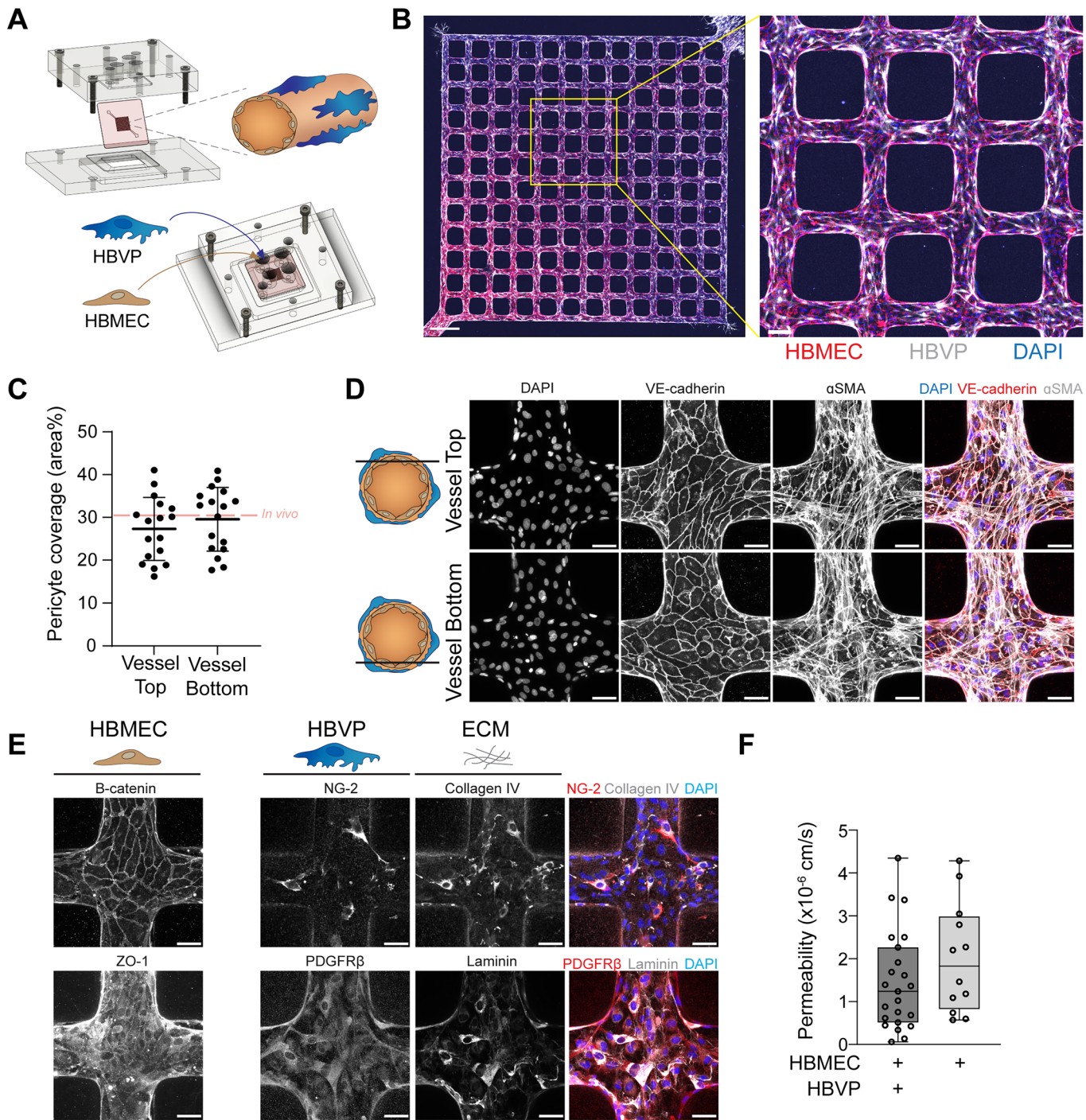

**Figure 1. Generation and characterization of a 3D brain microvasculature model.**

(A) Schematic depiction of the fabrication pieces and device set-up (top-left), including a representation of resultant endothelial–pericyte interactions (top-right), and the seeding method used to generate the 3D microvessel model (bottom). (B) IFA maximum z-projection of the full network of cellularized channels labeled with vWF for HBMEC (red), mCherry-expressing HBVP (white) and DAPI (blue) (left). Inset highlighting pericyte coverage of the microvessels (right). Scale bars: 500 μm and 100 μm (Inset). (C) Quantification of the percentage of endothelial microvessel area covered by pericytes on either the top or bottom microvessel surface in a z cross-sectional view. Red dashed line represents the estimated in vivo brain microvascular pericyte coverage. Data are presented as mean $+/-$ standard deviation ($n = 17$ devices, Mann–Whitney $U$ test). (D) IFA maximum z-projection of VE-cadherin, αSMA, and DAPI labeling on the top and bottom cross-sectional surfaces of microvessels (left). Merged maximum z-projections with VE-cadherin (red), αSMA (white), and DAPI (blue) (right). Scale bar: 50 μm. (E) IFA maximum z-projection of brain endothelial markers, ZO-1 and β-catenin, as well as, colocalization between brain pericyte markers, NG-2 and PDGFRβ (red), and extracellular matrix markers, laminin and collagen IV (gray) in the presence of DAPI staining (blue). Scale bar: 50 μm. (F) Apparent permeability of 70 kDa FITC-dextran in microvessels fabricated with HBMEC-only or a 5:1 HBMEC to HBVP ratio. Data presented as box and whisker plots displaying the median, 25th and 75th percentiles, and the minimum and maximum data points ($n = 25$ individual devices for HBMEC-only devices and 12 5:1 HBMEC to HBVP ratio devices, Mann–Whitney $U$ test). Source data are available online for this figure.

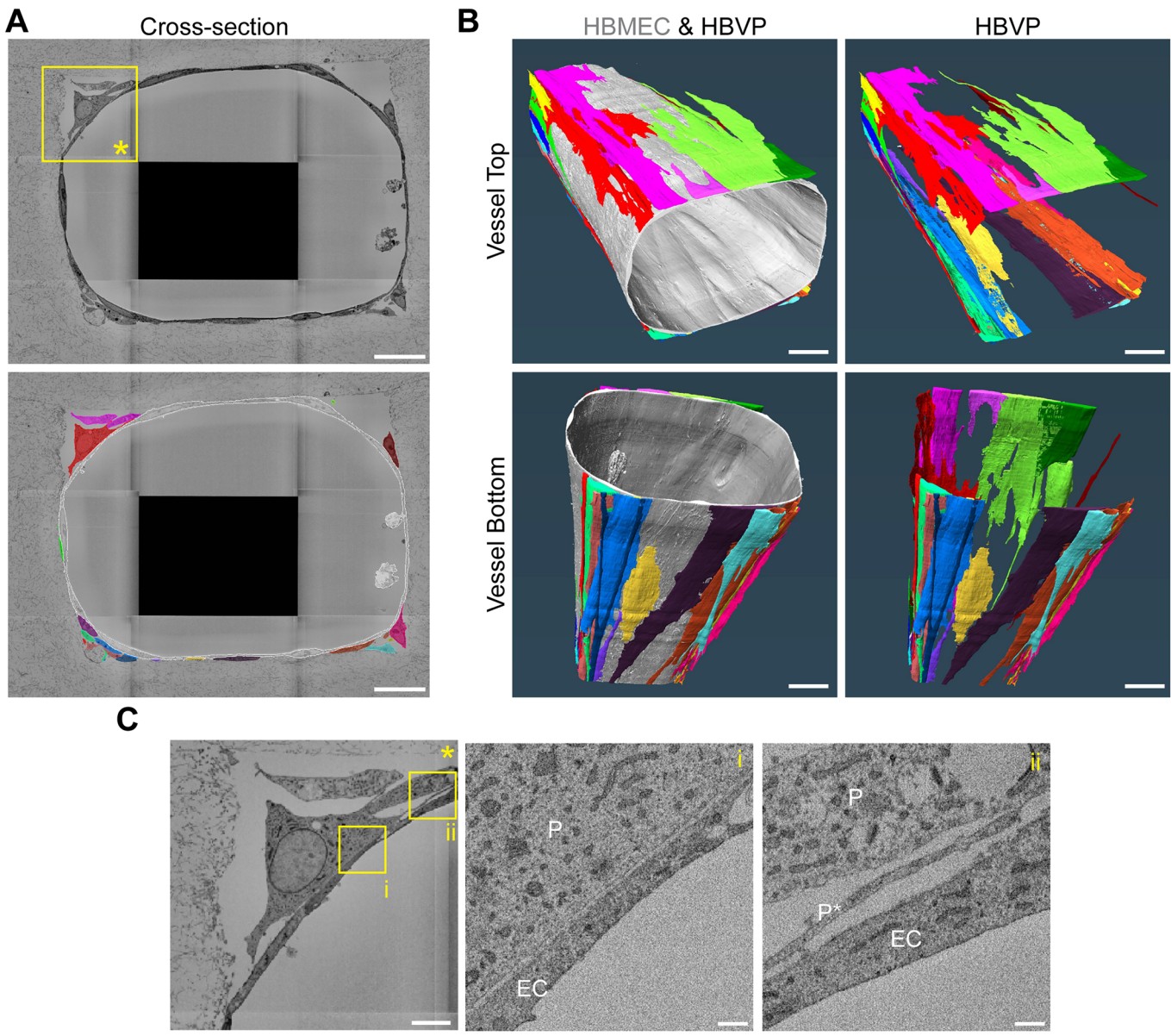

**Figure 2. Serial block-face scanning electron microscopy reveals endothelial cell-pericyte ultrastructural interactions.**

(A) 2D cross-section of the microvessel from the SBF-SEM volume (top) and the corresponding segmentation masks of the endothelium (white) and surrounding pericytes (colored). Scale bar: 20 µm. (B) 3D rendering of the segmented endothelium (gray) and the surrounding pericytes (colored) viewed from above or below. Scale bar: 10 µm. (C) Zoomed images of the region of interest (ROI) in (A). Insets show ROIs i and ii, where pericytes are denoted with "P", endothelial cells by "EC", and pericyte lamellae by "P*". Scale bars: 5 µm (left) and 500 nm (center and right). Source data are available online for this figure.

(Ornelas et al, 2021; Haley and Lawrence, 2017; Bonney et al, 2022). To characterize the spatial and ultrastructural organization of pericytes and endothelial cells in the model, we utilized Serial Block-Face Scanning Electron Microscopy (SBF-SEM) (Denk and Horstmann, 2004), a volume electron microscopy technique that provides nanoscale resolution of samples in a large field of view (Peddie et al, 2022). We imaged a 100-µm segment of a microvessel branch with a $15 \times 15 \times 50\ nm^3$ voxel size, followed by cellular segmentation and rendering. We characterized the spatial distribution and morphology of approximately 25 individual HBVP, distinguished from HBMEC by the presence of a highly granulated cytoplasm (Nahirney et al, 2016). Cross-sectional analysis and

segmentation of the microvessel revealed an ovoid microvessel morphology, and confirmed the presence of HBVP at the abluminal side of the HBMEC (Fig. 2A). HBVP acquired a classical bump-on-a-log morphology, with the soma being the thickest region of the cell, and their longitudinal axis aligned with the direction of flow (Fig. 2B). Most of the HBMEC-HBVP interface was composed of thin pericytic lamellae (100–300 nm thickness) that covered large areas of the endothelial surface, with multiple thin branches appearing mostly at the pericyte periphery, as described previously (Ornelas et al, 2021). Zoomed regions of HBVP-HBMEC cell interaction revealed that the membranes of both cell types were in close proximity (Fig. 2C). Brain pericytes constitute a

heterogeneous cell population depending on their location within the vascular branch. Considering the arteriole-sized diameter (100 μm) of the microvessels, the co-expression of αSMA, PDGFRβ, and NG-2 staining, and the bump-on-a-log morphology, HBVP within the model were reminiscent of ensheathing pericytes, commonly found on brain arterioles rather than capillary pericytes that appear not to express αSMA (Smyth et al, 2018; Grant et al, 2019). Overall, the 3D in vitro brain microvessel model recapitulated in vivo interactions between brain endothelial cells and pericytes, both at the tissue and ultrastructural level, that can next be utilized to model the role of pericytes in malaria pathogenesis.

## P. falciparum egress products increase vessel permeability but only cause minor changes in pericyte morphology

In vitro studies have shown that *P. falciparum* products released during iRBC parasite egress disrupt the endothelial barrier (Gillrie et al, 2007; Gallego-Delgado et al, 2016; Pal et al, 2016; Storm et al, 2019; Moxon et al, 2020). To characterize the potential impact of these products on pericyte function, we established a protocol to create media enriched for *P. falciparum*-iRBC egressed products, denoted from here on as iRBC-egress media. Briefly, tightly synchronized schizont stage-iRBC were purified and allowed to egress in vascular growth media, followed by recovery of the supernatant fraction containing *P. falciparum* soluble products at an estimated concentration of $5 \times 10^7$ egressed iRBC/mL (equivalent to simultaneous egress of parasites at a parasitemia of ~1%) (Fig. EV3A,B). We measured the endothelial disruptive properties of iRBC-egress media using the xCELLigence system, which provides real-time measurement of cell barrier integrity through impedance. Incubation with iRBC-egress media caused a significant, dose-dependent decrease in cell index, a measurement of monolayer barrier function normalized to untreated cells, reaching a maximum disruption at 18 h (Fig. EV3C). To elucidate if iRBC-egress media also induced barrier breakdown in the 3D brain microvessel model, we performed permeability assays in a simpler microvessel network which consists of a single 200-μm channel (Fig. EV2C). After perfusion and incubation with iRBC-egress media for 18 h, *P. falciparum* egress products accumulated on the walls of the microvessel channel and appeared as dark spots in brightfield suggesting the presence of parasitic hemozoin and food vacuoles (Fig. 3A). Microvessels perfused with iRBC-egress media presented a significant increase in 70 kDa FITC-dextran permeability across the entire vessel wall (median = $13.0 \times 10^{-6}$ cm/s), corresponding to a 4.3-fold increment compared to vessels perfused with media-only (median = $2.97 \times 10^{-6}$ cm/s) (Fig. 3A,B). Although the presence of inter-endothelial gaps was not observed, VE-cadherin staining revealed areas of discontinuity and thinning of adherens junctions in regions near the deposition of parasitic DAPI-positive DNA remnants (Fig. 3C). To better quantify changes in junction proteins upon barrier breakdown, we treated HBMEC monolayers with iRBC-egress media for 18 h and examined total protein levels of claudin-5 and VE-cadherin by western blotting. Upon iRBC-egress media incubation, a significant decrease in total claudin-5 protein levels was observed (Fig. 3D) while those of VE-cadherin remained unaltered (Fig. 3E), suggesting a role of tight junctional disruption in barrier breakdown by iRBC-egress media.

To determine whether increased permeability was associated with physical changes in pericytes, we first assessed pericyte coverage of the microvessel surface. In the presence of iRBC-egress media, pericytes maintained coverage of the endothelium to a similar degree of that observed in media-only-treated microvessels (Fig. 4A,B). Next, pericyte cellular morphology was examined at the ultrastructural level by SBF-SEM. In line with pericyte coverage analysis, HBVP were located around the entire microvessel perimeter (Fig. 4C,D) and maintained close spatial proximity to the endothelium in the presence of *P. falciparum*-iRBC ghosts, similar to control devices exposed to media-only (Fig. 4D). To investigate potential changes in HBVP shape, we isolated and analyzed each 3D-rendered HBVP (Figs. 4D,E and EV3D,E). Cellular features such as circularity and maximum width did not differ between conditions; however, iRBC-egress media-treated HBVP presented fewer branches, suggesting a response to iRBC-egress media (Fig. 4E). Altogether, our results show that *P. falciparum* egress products cause an increase in permeability of the 3D brain microvessel model, accompanied by disruption of endothelial claudin-5 and minor changes in pericyte morphology.

## P. falciparum-iRBC induce dysregulation of the angiopoietin–Tie axis

Next, we aimed to determine whether parasite egress products are a major driver of angiopoietin–Tie axis disruption characteristic of CM (Yeo et al, 2008; Lovegrove et al, 2009; Conroy et al, 2009, 2012; Higgins et al, 2016). A multiplexed Luminex assay targeting relevant proteins for endothelial–pericyte interaction was used to measure secreted protein concentrations in microvessel supernatants recovered after 18-h incubation with iRBC-egress media or media-only. Remarkably, we observed a substantial decline of Ang-1 secretion by HBVP following iRBC-egress media incubation (Fig. 5A). Consistent with recent studies on 2D HBMEC monolayers (Gomes et al, 2023), iRBC-egress media incubation did not result in increased Ang-2 secretion. Indeed, we quantified a significant decrease in Ang-2 secretion (Fig. 5A). Despite Ang-2 decrease, a significant increase in the Ang-2:Ang-1 ratio was found (Fig. 5A). In addition, we observed a decrease in 3D microvessel secretion or shedding of other vascular factors important for endothelial–pericyte interaction, such as Neural cadherin (N-cadherin) and Tissue Inhibitor of Metalloproteinase-1 (TIMP-1) (Fig. EV4A). Conversely, we detected no changes in soluble Vascular Endothelial Growth Factor (VEGF), Tie-2, Angiopoietin-like-4 (ANGPTL4), Platelet Derived Growth Factor-BB (PDGF-BB), Interleukin-8 (IL-8) or Chemokine Ligand-1 (CXCL-1) after 18-h incubation with iRBC-egress media (Fig. EV4A,B).

To validate whether changes in the angiopoietin–Tie axis occur physiologically following sequestration of *P. falciparum*-iRBC in the microvasculature, we perfused the 3D brain microvessels with schizont stage-iRBC for 30 min followed by a 10-min wash. After an 18-h incubation period, supernatants were again recovered and analyzed by Luminex. Similar results were obtained, with a reduction in both Ang-1 and Ang-2 secretion and an increase in the Ang-2:Ang-1 ratio (Fig. 5B). Altogether, these results suggest that iRBC hamper Ang-1 secretion by pericytes, revealing a previously unappreciated role of pericytes in the dysregulation of the angiopoietin–Tie axis in CM.

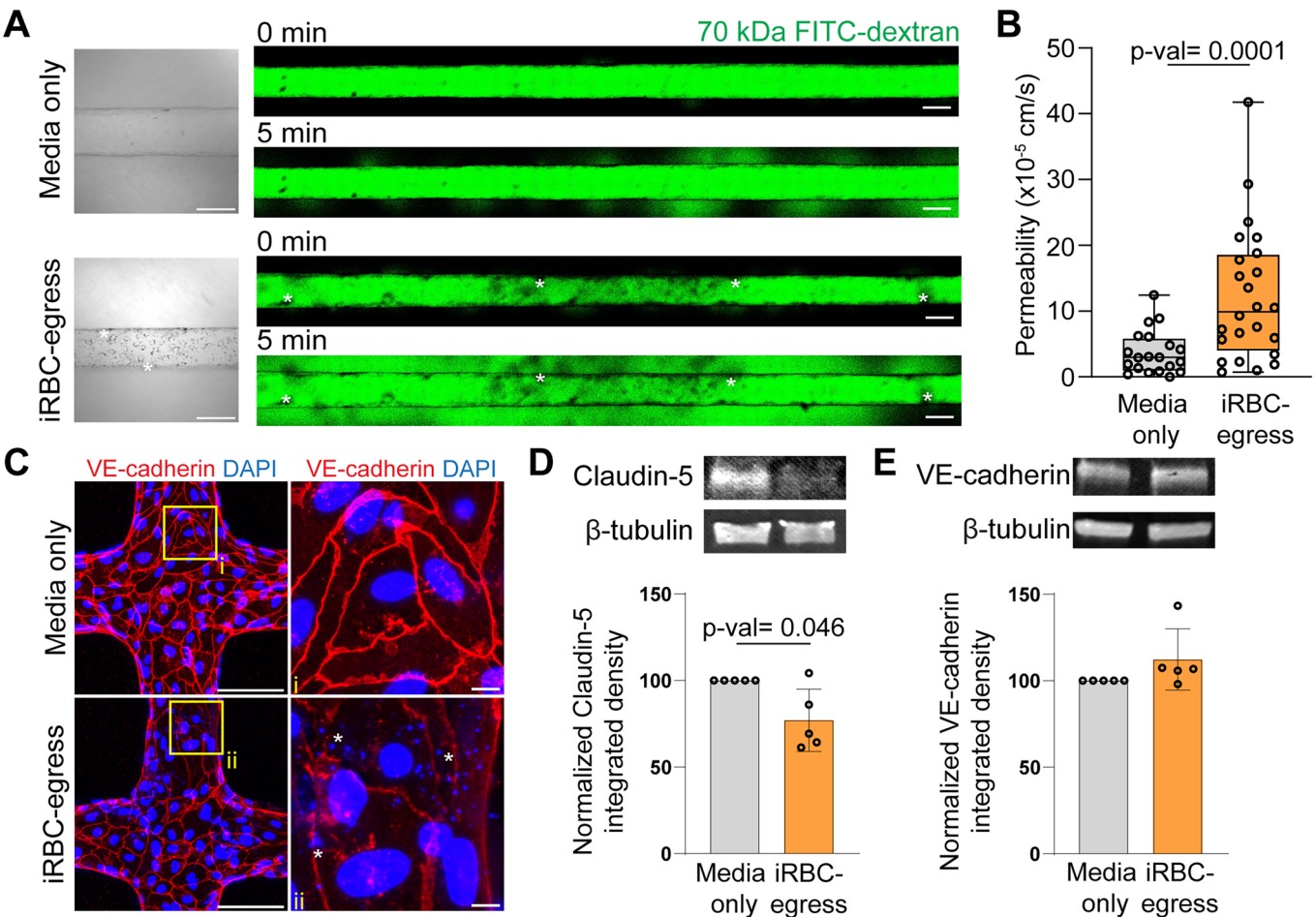

**Figure 3. iRBC egress products increase microvessel permeability through claudin-5 disruption.**

(A) Brightfield images of 3D brain microvessel channels after 18-h incubation with media-only or iRBC-egress media (left). Scale bar: 200 μm. Representative confocal images of 70 kDa FITC-dextran flux into the surrounding collagen 0 and 5 min after perfusion (right). Scale bar: 200 μm. Accumulated *P. falciparum*-iRBC egress material is represented with asterisks. (B) Apparent permeability of 70 kDa FITC-dextran in channels treated with media-only or iRBC-egress media for 18 h. ($n = 10$ individual devices incubated with media-only and 12 incubated with iRBC-egress media). (C) Immunofluorescence maximum z-projection of 3D brain microvessels exposed to media-only or iRBC-egress media stained with VE-cadherin (red) and DAPI (blue). Insets show ROIs of i and ii that display adherens junctions in the presence of cytoadhered DAPI-positive iRBC egress products represented by asterisks. Scale bar: 100 μm and 10 μm in insets. (D) Western blot of total claudin-5 protein levels following 18-h media-only or iRBC-egress media incubation of 2D HBMEC monolayers. Representative western blot images (top) and integrated density measurements normalized to the housekeeping protein, β-tubulin ($n = 5$ independent experiments). (E) Western blot of total VE-cadherin protein levels following 18-h media-only or iRBC-egress media incubation of 2D HBMEC monolayers. Representative western blot images (top) and integrated density measurements normalized to the housekeeping protein, β-tubulin. ($n = 5$ independent experiments). In (B), data are presented as box and whisker plots displaying the median, 25th and 75th percentiles and the minimum and maximum data points (Mann–Whitney $U$ test). In (D, E), data are presented as bar charts displaying the mean $+/-$ standard deviation (one-sample $T$ test). Source data are available online for this figure.

## Therapeutic targeting of the angiopoietin–Tie axis partially protects against endothelial barrier breakdown by iRBC-egress media

Currently, there are no available host-targeted adjunctive therapies to restore brain vascular barrier function during CM. Given the key role of Ang-1 in promoting vascular quiescence and endothelial barrier formation, we first investigated whether recombinant Ang-1 (rAng-1) supplementation could protect against the barrier-disruptive effects of iRBC-egress media. First, we measured barrier disruption by xCELLigence in a 2D in vitro HBMEC culture in the absence of HBVP. rAng-1 pre-treatment experiments were done in the absence of serum, an external source of Ang-1 (Fig. EV4C).

An 18-h pre-treatment of HBMEC with rAng-1 prior to exposure to iRBC-egress media partially protected against parasite-mediated endothelial barrier disruption, while a 1-h pre-treatment did not display a protective effect (Fig. EV5A–F). Indeed, the 18-h pre-treatment with rAng-1 conferred a 40% barrier breakdown protection after normalization to the untreated endothelial condition (Fig. EV5F). Next, we tested the protective capacity of rAng-1 against iRBC-egress products in the 3D microvessel model (Fig. 6A), also in the absence of serum, which non-significantly increased the baseline permeability of resting devices (Fig. EV5G). Microvessels exposed to iRBC-egress media (median = $9.90 \times 10^{-6}$ cm/s) presented a normalized 2.3 increase in permeability compared to a media-only control (median = $4.39 \times 10^{-6}$ cm/s), while microvessels exposed to

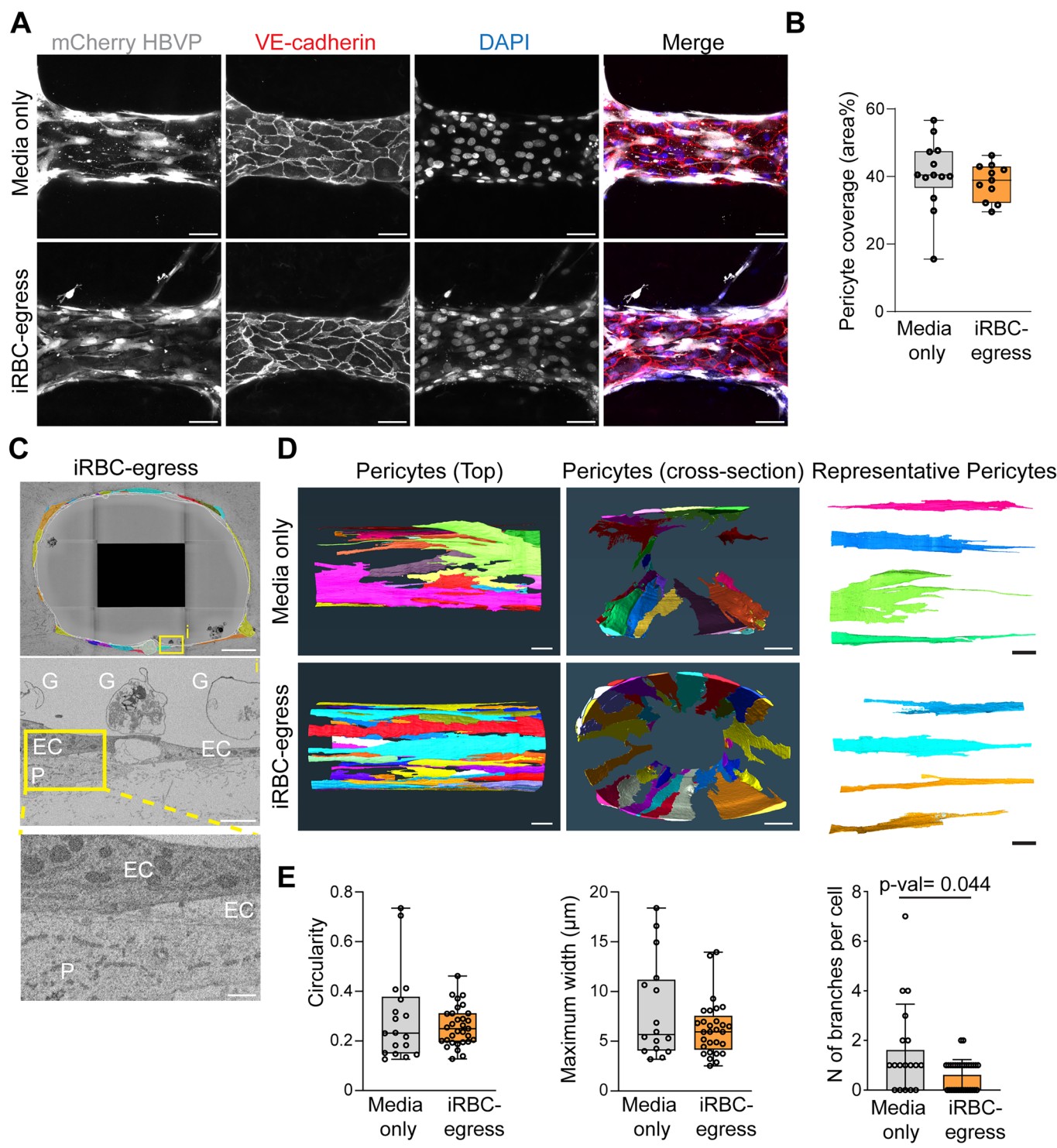

iRBC-egress media and pre-treated with Ang-1 (median = 9.13 × $10^{-6}$ cm/s) presented a normalized 1.5-fold increase in permeability compared to microvessels pre-treated with Ang-1 (median = 6.17 × $10^{-6}$ cm/s), representing a 36% protection against microvessel leakage after exposure to iRBC-egress media (Fig. 6B,C). Taken together, pre-treatment with rAng-1 partially protects against the microvascular permeability caused by iRBC egress products, highlighting the

importance that a decrease in Ang-1 secretion could have in exacerbating CM-induced vascular disruption, as well as, showing a potential for vascular restorative pathways that target the angiopoietin–Tie axis in CM treatment.

During angiogenesis, vascular endothelial-protein tyrosine phosphatase (VE-PTP) regulates Tie-2 activity by dephosphorylation (Fachinger et al, 1999). AKB-9778, a pharmacological inhibitor

Figure 4.   Pericytes maintain endothelial coverage while displaying minor changes in morphology in the presence of iRBC egress products.

(A) Immunofluorescence maximum z-projection of the bottom surface of 3D brain microvessels exposed to media-only or iRBC-egress media stained with VE-cadherin (red) and DAPI (blue), with HBVPs visualized by expression of mCherry (white). Scale bar: 50 μm. (B) Quantification of the percentage of endothelial microvessel area covered by pericytes on the bottom surface of 3D brain microvessels exposed to media-only or iRBC-egress media (n = 4 individual devices incubated with media-only and 3 incubated with iRBC-egress media). (C) SBF-SEM cross-sectional image (top) of iRBC-egress exposed microvessels with corresponding segmentation masks of the endothelium (white) and surrounding pericytes (colored). Inset (middle) and zoomed ROI (bottom) display endothelial and pericyte ultrastructural contact in the presence of cytoadhered P. falciparum-iRBC ghosts. Pericytes are denoted as "P", endothelial cells by "EC" and iRBC ghosts by "G". Scale bar: 20 μm (top), 2 μm (middle) and 500 nm in inset (bottom). (D) Microvessel top and cross-sectional view of 3D-rendered pericytes segmented from microvessels treated with media-only or iRBC-egress media and imaged with SBF-SEM (left). Scale bar: 10 μm. Representative segmented pericytes for morphology analysis (right). Scale bar: 10 μm. (E) Circularity, maximum width, and number (N) of branches analysis (n = 18 or 31 segmented pericytes treated with media-only or iRBC-egress media, respectively). Data information: In (B, E) data are presented as box and whisker plots displaying the median, 25th and 75th percentiles and the minimum and maximum data points (Mann–Whitney U test). Source data are available online for this figure.

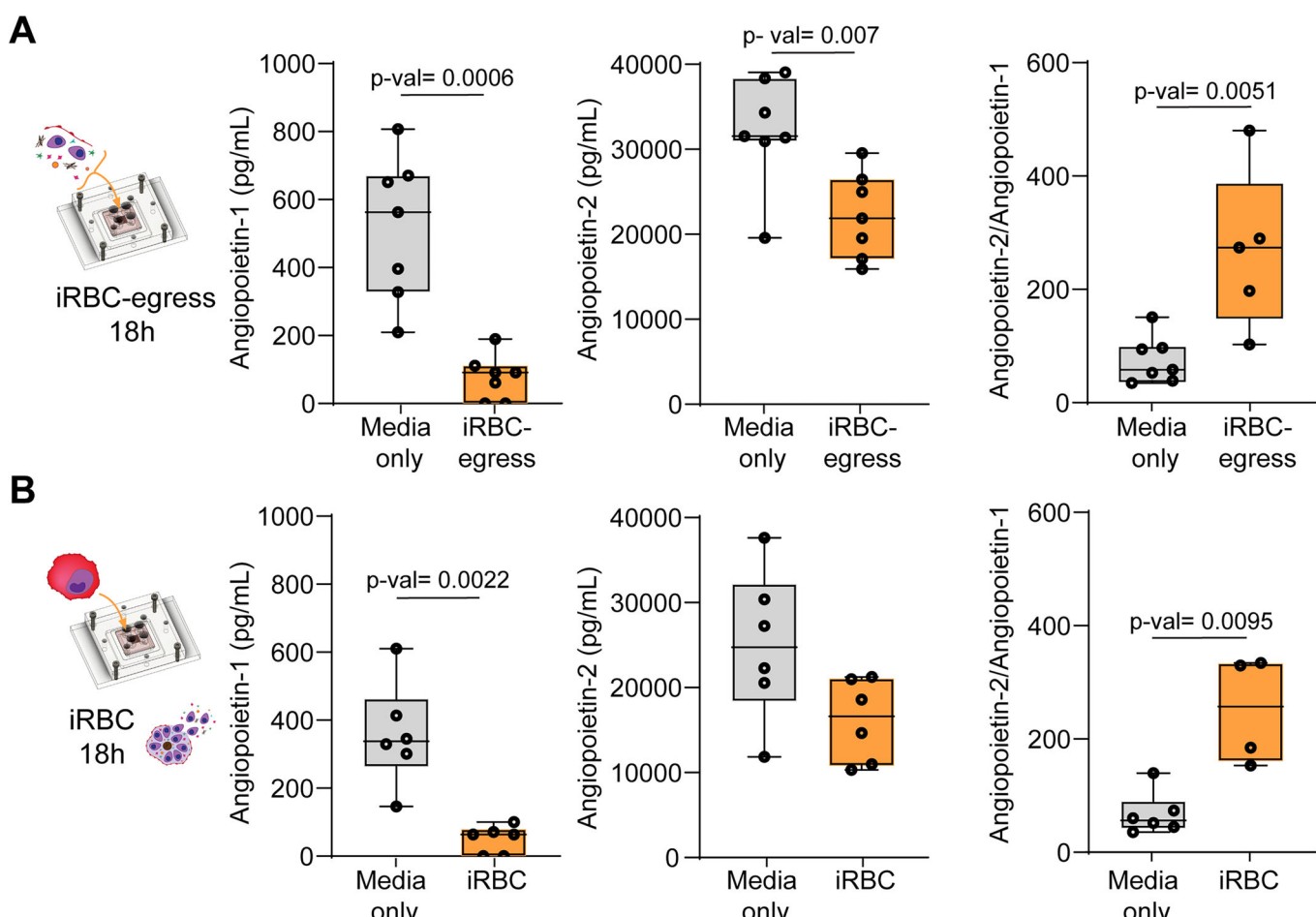

Figure 5.   iRBC egress products induce angiopoietin–Tie axis dysregulation by altering pericyte secretion of Ang-1.

(A, B) Ang-1, Ang-2 concentrations, and Ang-2: Ang-1 ratio measured from 3D brain microvessels treated with media-only, iRBC-egress media (A) or P. falciparum-iRBC (B) for 18-h (n = 7 pooled supernatants from two to three devices each for iRBC-egress media incubation and 6 pooled supernatants from 2 to 3 devices each for iRBC incubation). Concentrations were normalized by subtracting baseline protein concentrations measured from freshly made media (Fig. EV4C). Data information: Box and whisker plots display the median, 25th and 75th percentiles and the minimum and maximum data points (Mann–Whitney U test) In (A, B), data points in which Ang-1 concentrations were below the limit of detection were removed to calculate the Ang-2: Ang-1 ratio. Source data are available online for this figure.

of VE-PTP activity, has been shown to specifically activate the angiopoietin–Tie axis to promote endothelial barrier function in the presence of various mediators of vascular disruption and inflammation (Frye et al, 2015; Shen et al, 2014). Therefore, AKB-9778 is currently undergoing phase II clinical trials as a therapeutic treatment for retinal vascular diseases characterized by

angiopoietin–Tie axis dysregulation such as diabetic macular edema and retinopathy (Campochiaro et al, 2016) and diabetic hypertension (Siragusa et al, 2021). As the retinal and brain vasculature share similar embryological origin, anatomy, cell composition and response during CM (MacCormick et al, 2014), we tested the potential of AKB-9778 as a therapeutic option to

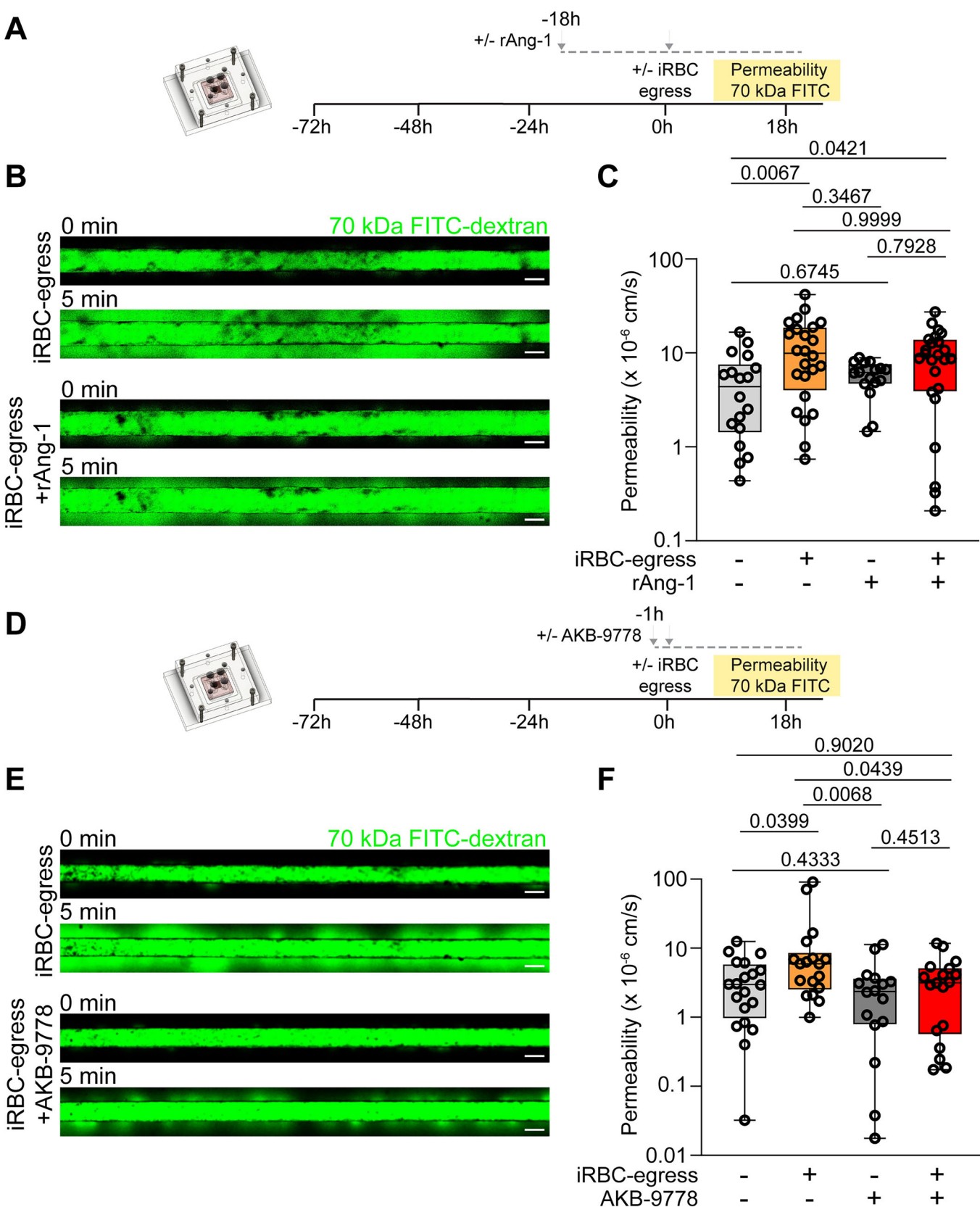

◄ **Figure 6.** Therapeutic activation of the angiopoietin–Tie axis partially protects against increased permeability induced by iRBC egress products.

(A) Experimental outline of the rAng-1 pre-treatment permeability experiment on 3D brain microvessels. (B) Confocal image of the 70 kDa FITC-dextran flux into the surrounding collagen at 0 and 5 min after dextran perfusion. Images correspond to a representative iRBC-egress media-treated microvessel pre-treated with either media-only or rAng-1 for 18 h. Representative image of an iRBC-egress media-treated microvessel pre-treated with media-only reused from Fig. 3A. Scale bars: 200 μm. (C) Apparent permeability of 70 kDa FITC-dextran in microvessels pre-treated with serum-free media $+/-$rAng-1 followed by 18-h media-only or iRBC-egress media treatment ($n = 9$ individual devices for serum-free media-only incubation, 12 for iRBC-egress media incubation, 8 for serum-free media+rAng-1 incubation, and 12 for iRBC-egress media+rAng-1 incubation). (D) Experimental outline of the AKB-9778 pre-treatment permeability experiment on 3D brain microvessels. (E) Confocal image of the 70 kDa FITC-dextran flux into the surrounding collagen at 0 and 5 min after dextran perfusion. Images correspond to a representative iRBC-egress media pre-treated with either media-only or AKB-9778 for 1 h. Scale bars: 200 μm. (F) Apparent permeability of 70 kDa FITC-dextran in microvessels pre-treated with or without AKB-9778 followed by 18-h media-only or iRBC egress media treatment ($n = 10$ individual devices for media-only incubation, 9 for iRBC-egress media incubation, 8 for AKB-9778 incubation and 9 for iRBC-egress media+AKB-9778 incubation). Data information: Box and whisker plots display the median, 25th and 75th percentiles and the minimum and maximum data points in (C, F) (Kruskal–Wallis test corrected for multiple comparisons using the Benjamini, Krieger and Yekutieli method). Source data are available online for this figure.

counteract CM. In a 2D in vitro HBMEC culture, both 1- and 18-h pre-treatment with AKB-9778 resulted in a partial protection against parasite-mediated endothelial barrier disruption of 56% and 43%, respectively (Fig. EV5H–M). Given the ability of AKB-9778 to protect against barrier disruption in 2D monolayers, we next validated its rapid protective activity in the 3D microvessel model (Fig. 6D). Microvessels pre-treated for 1 h with AKB-9778 and exposed to iRBC-egress media (median = 3.1 $5 \times 10^{-6}$ cm/s) presented a non-significant 1.3-fold increase in permeability compared to AKB-9778 pre-treated microvessels (median = $2.35 \times 10^{-6}$ cm/s), while untreated microvessels exposed to iRBC-egress media (median = $5.90 \times 10^{-6}$ cm/s) presented a significant twofold increase in permeability compared to a media-only control (median = $2.97 \times 10^{-6}$ cm/s), representing a 35% protection against microvessel leakage after exposure to iRBC-egress media (Fig. 6E,F). Furthermore, a significant decrease in permeability was observed in iRBC-egress media-exposed devices pre-treated with AKB-9778. The significant but partial vascular barrier protective ability of AKB-9778 further highlights a crucial role of the angiopoietin–Tie axis in CM pathogenesis, as well as, suggests VE-PTP inhibition as a potential adjuvant therapeutic option for CM treatment.

## Discussion

Disruption of the angiopoietin–Tie axis is a common feature in CM patients, often associated with disease severity and fatality (Lovegrove et al, 2009; Conroy et al, 2012). This dysregulation is also found in pediatric and adult severe malaria patients without cerebral complications (Yeo et al, 2008; Conroy et al, 2009) or in placental malaria (Tran et al, 2021). Central to this pathway in the brain microvasculature are pericytes, an Ang-1-secreting cell type that is located at the interface between the microvasculature and brain parenchyma, yet largely understudied in the context of CM. Here, we developed a 3D microfluidic brain microvessel model that recapitulates the minimal functional brain microvascular unit to recreate the angiopoietin–Tie axis, including both primary brain microvascular endothelial cells and pericytes. Exposure of the 3D brain microvessels to *P. falciparum*-iRBC egress products caused an increase in vascular permeability, accompanied by decreased expression of the tight junction protein claudin-5 and a decrease in pericyte secretion of Ang-1. Furthermore, we showed that pre-treatment with rAng-1 or the Tie-2 activator, ABK-9778, could partially protect against endothelial barrier breakdown. Taken

together, these results suggest that the cessation of Ang-1 release upon exposure to iRBC-egress products is an important mechanism in microvessel disruption.

In vitro microfluidics-based brain vascular models have been recently utilized in the study of several infectious diseases, including those caused by malaria (Howard et al, 2023; Bernabeu et al, 2019), neurotropic viruses (Zhang et al, 2025; Wang et al, 2024) and fungi (Kim et al, 2021). However, most of these models do not include pericytes or lack detailed characterization of brain endothelial–pericyte interactions. Here, we fabricated an in vitro model of the human brain microvasculature by co-seeding primary brain endothelial cells and pericytes through the microfluidic network. Although vessels simultaneously seeded with a 3:1 endothelial cell:pericyte ratio were not viable, a 5:1 ratio reproduced the high pericyte coverage found in the microvasculature of the human brain. Furthermore, the presence of brain pericytes at the 5:1 ratio did not prevent the formation of an intact endothelial microvessel network with continuous labeling of endothelial adherens and tight junctional markers and decreased permeability. SBF-SEM microscopy showed that both cell types sit at nanoscale distance, with the presence of thin lamellae extending from the pericyte soma that cover large areas of the endothelium. Despite these advancements, the ratio of endothelial cells to pericytes within the model was still higher than the 3:1 to 1:1 endothelial:pericyte ratio reported in vivo (Sims, 1986; Mathiisen et al, 2010) Therefore, future improvements to the model could include concurrent addition of pericytes to the collagen bulk to increase the number of pericytes within the model without inducing extensive microvessel contraction. Overall, the 3D brain microvessel model recapitulates pericyte–endothelial interactions and coverage found in human brain vasculature making it suitable to study the role of pericytes in CM pathogenesis.

*P. falciparum* products released during blood stage egress have been shown to possess barrier breakdown capabilities on 2D endothelial monolayers grown on plastic. Proposed barrier-disruptive *P. falciparum*-iRBC egress products include merozoite proteins, histones, heme, hemozoin, and *P. falciparum* histidine-rich protein 2 (Gomes et al, 2023; Gillrie et al, 2007; Gallego-Delgado et al, 2016; Pal et al, 2016; Storm et al, 2019; Moxon et al, 2020; Zuniga et al, 2022). Our studies in 2D revealed, for the first time in HBMEC, that exposure to iRBC-egress products induces barrier breakdown as a consequence of decreased protein expression of the tight junctional marker claudin-5, in agreement with the well-documented role of claudin-5 as a key regulator of

blood–brain barrier breakdown in in vivo models of multiple sclerosis (Argaw et al, 2009; Paul et al, 2013), traumatic brain injury (Campbell et al, 2012) and depression (Sun et al, 2024). Our results further show that vascular breakdown was also found in the bioengineered 3D brain microvascular model, in the presence of mechanical cues known to improve microvascular integrity, including the presence of physiological flow and tissue stiffness. Pericytes also play an important role in maintaining vascular homeostasis and therefore loss of pericytes from the cerebral vasculature leads to increased permeability in mouse models (Armulik et al, 2010). Furthermore, dysregulation of cerebral pericytes and detachment from the vasculature have been observed in human cerebral vasculature pathologies, including stroke and Alzheimer's disease (Fernández-Klett et al, 2013; Sengillo et al, 2013). In our study, we did not observe gross differences in pericyte morphology or loss of vascular coverage upon treatment with iRBC-egress media. Nevertheless, pericytes from iRBC-egress media-treated 3D microvessels did display subtle morphological changes, having significantly less branches. The relevance of these minor ultrastructural modifications in vascular barrier dysfunction could imply molecular mechanisms of disease. Luminex quantification upon stimulation with iRBC-egress media in the 3D brain microvessel model revealed a significant reduction in N-cadherin shedding, suggesting reduced functional endothelial–pericyte interaction upon short-term incubation with *P. falciparum*-egress products. Future studies should explore whether exposure to *P. falciparum* for longer time points recapitulates pericyte phenotypes observed in postmortem samples of cerebral malaria patients, such as vacuolization (Pongponratn et al, 2003) or vascular pericyte loss (Barrera et al, 2018). Nevertheless, our results point towards a loss of integrity at the endothelial and pericyte interface.

Ang-1 and Ang-2 have been considered in the past as potential biomarkers of CM (de Jong et al, 2016). Ang-1 has excellent predictive power to distinguish malaria severity scores (Conroy et al, 2009) or malaria from other central nervous system febrile diseases (Conroy et al, 2010). Nevertheless, the underlying mechanism of Ang-1 decrease in CM remains unknown, as previous in vitro studies overlooked the effect that iRBC could exert on brain pericytes. To our knowledge, our study is the first one to report that *P. falciparum* egress products released upon iRBC egress halt Ang-1 protein secretion by pericytes. This is in line with a recent single-cell RNA sequencing analysis from our group showing decreased transcriptional expression of Ang-1 in pericytes and Tie-2 (TEK) in endothelial cells following iRBC-egress media incubation (Piatti et al, 2025). Nevertheless, an unresolved question remains whether pericyte dysregulation is directly mediated by parasite egress products crossing the vascular barrier, or through disrupted endothelial–pericyte crosstalk. Furthermore, future studies are necessary to isolate and identify which of the multiple iRBC egress components are responsible for altered signaling pathways in the microvasculature. Although most of them have shown strong disruptive effects when added separately to 2D endothelial monolayers, it remains unclear whether they will exert the same degree of damage once isolated, as observed following iRBC-egress media incubation in this 3D model with pericyte co-culture and enhanced barrier properties. Furthermore, other cell types not included in the model could further contribute to the decreased Ang-1 levels observed in CM patients. For example, severe thrombocytopenia is strongly

associated with cerebral complications during *P. falciparum* infections (Sahu et al, 2022) and could play a role as platelets represent another major source of serum Ang-1 (Nachman and Rafii, 2008). In light of this, it is likely that both the effect of parasite egress products on perivascular cells and platelet depletion could work in concert to further contribute to a widespread decrease in Ang-1 levels. Nevertheless, our results highlight brain pericytes as a new contributor in the development of CM through dysregulation in the secretion of Ang-1.

Clinical studies have shown a lack of correlation between Ang-2 levels and *P. falciparum*-iRBC microvascular obstruction in the rectal microvasculature (Hanson et al, 2015) or iRBC sequestration in brain postmortem samples (Prapansilp et al, 2013). Similarly, our study confirms that *P. falciparum*-iRBC sequestration in 3D brain microvessel models does not directly cause Ang-2 secretion by endothelial cells. This result is in concordance with a recent study that also showed that Ang-2 is not secreted in response to *P. falciparum* lysate products (Gomes et al, 2023). Instead, other studies have shown Ang-2 release as a result of exposure to TNFα, thrombin or hypoxia (Pichiule et al, 2004; Fiedler et al, 2004), conditions that have also been associated with CM pathogenesis. Therefore, our study is in agreement with previous reports suggesting that *P. falciparum* sequestration does not play a direct role in the secretion of Ang-2, which instead is more likely a consequence of the activation of host pro-inflammatory factors.

Modulation of the angiopoietin–Tie axis as a therapeutic target against CM has long been of interest (Higgins et al, 2016) due to its role in strengthening endothelial tight junctions, as well as upregulating anti-apoptotic and anti-inflammatory pathways (Saharinen et al, 2017). Previously, administration of a recombinant Ang-1 tetramer protein with improved stability, BowAng1, has shown improved outcomes in a rodent experimental CM model even after onset of disease, preserving blood–brain barrier integrity, and leading to a significantly increased survival rate (Higgins et al, 2016). However, translation of therapeutics identified in rodent experimental CM models to the clinic have proven challenging. For example, inhalation of nitric oxide, an activator of the angiopoietin–Tie axis (Zacharek et al, 2006), was shown to be successful in rodent experimental CM models (Serghides et al, 2011) but showed no improvement in a human clinical trial (Hawkes et al, 2015). Therefore, we believe that the future combination of experimental approaches using 3D bioengineered microvessels with mouse *P. berghei* models would enhance translation and development of adjunctive therapies, improving the clinical care of CM patients. Interestingly, we showed that pre-treatment of human microvessels with recombinant Ang-1 partially protected against barrier breakdown, highlighting the importance of the angiopoietin–Tie axis in iRBC-egress mediated disruption. While the concentration of recombinant Ang-1 utilized in this study was supraphysiological compared to peripheral blood samples, it is quite likely that peripheral blood concentrations underestimate local concentrations at the endothelial–pericyte interface. In addition, in our model, recombinant Ang-1 protection was only achieved after a pre-treatment of 18-h, suggesting that downstream signaling through the Tie2 receptor, such as PI3K/Akt signaling, is required (Augustin et al, 2009).

Future interventions targeting the angiopoietin–Tie axis should focus on approaches that quickly reverse microvascular dysfunction after exposure to *P. falciparum*. Pre-treatment with AKB-9778

resulted in rapid partial protection against iRBC egress product-mediated breakdown, highlighting the potential of the angiopoietin–Tie axis as an adjuvant therapeutic target for CM treatment. Furthermore, AKB-9778-mediated activation of Tie-2 could provide an additional benefit in combating CM pathogenesis as it has also been shown to block the binding of leukocytes to the vascular wall (Frye et al, 2015), with recent reports describing their accumulation in the brain vasculature of CM patients (Riggle et al, 2020). As AKB-9778 has been well-tolerated during Phase II clinical trials for the treatment of diabetic retinopathy (Campochiaro et al, 2016), future studies examining its ability to reverse vascular dysfunction in malaria patients would provide a deeper understanding of its use to treat CM. Altogether, our results underline our model as a valuable platform to identify potential therapeutics and, in line with previous in vivo mouse models, highlight the potential of compounds targeting the angiopoietin–Tie axis as adjunctive CM therapies.

To note, our study has several limitations, including the low throughput of the model or challenges in growing the devices for long periods of time. Previous studies have maintained a month-long culture of hydrogel-based in vitro microvasculature models by introduction of a constant laminar or intermittent flow, allowing for the study of long-term vascular pathology and recovery (Qiu et al, 2018; Cherubini et al, 2023). Thus, the addition of constant or intermittent flow to the model could aid in identification of long-term disruptive mechanisms involved in CM, such as associated pericyte loss or vascular remodeling. Furthermore, our devices currently represent larger microvessels of the brain given the 100 µm diameter of our model and α-SMA expression on pericytes. Therefore, future studies could exploit 2-photon laser ablation or bioengineered self-assembled models (Arakawa et al, 2020; Hajal et al, 2022) to generate brain capillary-sized microvessels (5–10 µm), a major pathogenic site in CM. Nevertheless, this study establishes a key role of pericytes on microvessels that are similar in size to arterioles or post-capillary venules, regions of the brain vascular tree that are equally damaged in CM patients (Barrera et al, 2018). Finally, an additional limitation of our model is the absence of other cells that could play an important role in maintaining the angiopoietin–Tie axis and vascular barrier function. For example, Ang-1 secretion has been described in other brain and blood cell types, such as smooth muscle cells (Sundberg et al, 2002; Nishishita and Lin, 2004), platelets, and more recently, neurons and astrocytes (Wei et al, 2021). In addition, cell types such as astrocytes have been shown to further aid in strengthening the barrier properties of in vitro models, as the permeability displayed here remained supraphysiological to values observed in vivo (DeStefano et al, 2018). Yet, one of the major advantages of in vitro bioengineered models is the opportunity to sequentially introduce different components that, independently or collectively, could play a role in a complex and multifaceted disease such as CM. Subsequent iterations of the model presented here could introduce other cell types that produce Ang-1 or incorporate pro-inflammatory cytokines or thrombin, to concurrently model functional consequences of increased secretion of Ang-2 by endothelial cells.

In summary, our study highlights the role that pericytes could have in the development of CM pathogenesis by showing for the first time that egress of *P. falciparum*-iRBC interrupts Ang-1 secretion by this cell type. In addition, our findings confirm that activation of the angiopoietin–Tie axis partially protects against increased vascular permeability mediated by *P. falciparum* egress products, providing further evidence that interventions involving this pathway could be part of a multi-targeted adjunctive therapy against CM.

# Methods

## Reagents and tools table

| Reagent/resource | Reference or source | Identifier or catalog number |
|---|---|---|
| **Experimental models** | | |
| HB3var03 (*P. falciparum*) | Avril et al, 2016; Dr. Joe Smith, Center of Global Infectious Disease Research, Seattle; Dr. Maria Bernabeu, EMBL Barcelona, Barcelona | N/A |
| Human brain microvascular endothelial cells (*H. sapiens*) | Cell Systems | Cat #ACBRI 376 |
| Human brain vascular pericytes (*H. sapiens*) | ScienCell | Cat #1200 |
| **Antibodies** | | |
| Rabbit anti-αSMA | Abcam | Cat #EPR5368 |
| Mouse anti-vWF | Santa Cruz | Cat #sc-365712 |
| Mouse anti-ZO-1 | Invitrogen | Cat #10017242 |
| Rabbit anti-β-catenin | Cell Signaling | Cat # 9587S |
| Mouse anti-PDGFRβ | Abcam | Cat #ab69506 |
| Mouse anti-NG2 | Invitrogen | Cat #10424493 |
| Rabbit anti-Laminin | Abcam | Cat #ab11575 |
| Rabbit anti-Collagen IV | Abcam | Cat #ab6586 |
| Mouse anti-VE-cadherin | Santa Cruz | Cat #sc-52751 |
| Mouse anti-CD31 | Abcam | Cat #ab215911 |
| Phalloidin-647 | Invitrogen | Cat #A22287 |
| Rabbit anti-VE-cadherin | Cell Signaling | Cat #D87F2 |
| Rabbit anti-Claudin-5 | Cell Signaling | Cat #E8F3D |
| Mouse anti-β-tubulin | Invitrogen | Cat #MA5-16308 |
| Goat anti-mouse Alexa Fluor 488 | Invitrogen | Cat #A-11001 |
| Goat anti-mouse Alexa Fluor 594 | Invitrogen | Cat #A-11005 |
| Goat anti-mouse Alexa Fluor 647 | Invitrogen | Cat #A-21235 |
| Goat anti-rabbit Alexa Fluor 488 | Invitrogen | Cat #A-11008 |
| Goat anti-rabbit Alexa Fluor 594 | Invitrogen | Cat #A-11012 |
| Goat anti-rabbit Alexa Fluor 647 | Invitrogen | Cat #A-21244 |

| Reagent/resource | Reference or source | Identifier or catalog number |
|---|---|---|
| DAPI | Invitrogen | Cat #D21490 |
| IRDye goat anti-mouse 680RD | LI-COR Biotech | Cat #926-68070 |
| IRDye goat anti-rabbit 800CW | LI-COR Biotech | Cat #926-32211 |
| **Chemicals, enzymes, and other reagents** | | |
| Compound 2 | Koussis et al, 2020 | Dr. Michael Blackman, The Francis Crick Institute, London |
| Recombinant angiopoietin-1 | R&D Systems | Cat #923-AB-025 |
| AKB-9778 | MedChemExpress | Cat #HY-1009041-1MG |
| Pierce RIPA buffer | Invitrogen | Cat #89900 |
| 1X Protease inhibitor | Invitrogen | Cat #A32963 |
| BCA assay | Invitrogen | Cat #23227 |
| Mini-PROTEAN TGX gels | Bio-Rad | Cat #4561034 |
| Nitrocellulose membranes | Bio-Rad | Cat #1704159 |
| 96 well PET E-plates | Agilent | Cat #300600910 |
| Poly-ʟ-Lysine | Sigma | Cat #P8920 |
| Pericyte Media | ScienCell | Cat#1201 |
| Endothelial growth media-2MV | Lonza | Cat #CC-3202 |
| Astrocyte Growth Supplement | ScienCell | Cat #1852 |
| Pericyte Growth Supplement | ScienCell | Cat #1252 |
| µ-Slide 8-well high glass bottom chambered coverslips | IBIDI | Cat #80807 |
| 70 kDa FITC-dextran | Sigma | Cat #60842-46-8 |
| Odyssey One-Color Protein Molecular Weight Marker | LI-COR Biotech | Cat #928-40000 |
| Precision Plus Protein Western C Blotting Standard | Bio-Rad | Cat #1610376 |
| Human Angiopoietin-1 DuoSet ELISA kit | R&D Systems | Cat #DY923 |
| Human Angiopoietin-2 DuoSet ELISA kit | R&D Systems | Cat #DY623 |
| **Software** | | |
| Amira | https://www.thermofisher.com/es/es/home/electron-microscopy/products/software-em-3d-vis/amira-software.html | |
| GraphPad Prism10.3.0 | https://www.graphpad.com | |

| Reagent/resource | Reference or source | Identifier or catalog number |
|---|---|---|
| ImageJ | https://imagej.nih.gov/ij/index.html | |
| SBEMimage | Titze et al, 2018 | |
| **Other** | | |
| PHD 2000 Syringe Pump | Harvard Apparatus | |
| PELCO Biowave Pro + | Ted Pella | |
| EM UC7 Microtome | Leica Microsystems | |
| Trans-blot Turbo transfer machine | Bio-Rad | |
| LI-COR Odyssey M Scanner | LI-COR Biotech | |
| Human Luminex Discovery Assay | R&D Systems | |
| Luminex 100/200 | Luminex Corporation | |
| xCELLigence RTCA SP reader | Agilent | |
| Zeiss Gemini2 | Zeiss | |
| Zeiss LSM 980 Airyscan2 | Zeiss | |

## Methods and protocols

### Parasite lines

HB3var03 variant *P. falciparum* parasites (dual ICAM-1 and EPCR binding), regularly panned and monitored for appropriate PfEMP1 expression, were cultured in human B+ erythrocytes in RPMI 1640 medium (GIBCO) supplemented with 10% human type AB-positive plasma, 5 mM glucose, 0.4 mM hypoxanthine, 26.8 mM sodium bicarbonate and 1.5 g/L gentamicin. Parasites were grown in sealed top T75 flasks in a gas mixture of 90% $N_2$, 5% $CO_2$ and 1% $O_2$. To maintain synchronous cultures, parasites were synchronized twice a week in their ring stage with 5% sorbitol and once a week in their trophozoite stage with 40 mg/mL gelaspan (Braun).

### P. falciparum-iRBC egress media preparation

Late-stage *P. falciparum*-iRBC (schizonts, parasitemia of 5–10%), synchronized in a 6-h window, were purified by use of 40 mg/mL gelaspan gradient separation to a final purity of >60% parasitemia. The enriched *P. falciparum*-iRBC were then placed in complete RPMI media containing 1 µM compound 2, a reversible PKG inhibitor that inhibits iRBC egress (kindly donated by Michael Blackman, The Francis Crick Institute) (Koussis et al, 2020), at a concentration of 50 million *P. falciparum*-iRBC/mL. After 5 h, media containing compound 2 was removed and the parasites were resuspended at a concentration of 100 million *P. falciparum*-iRBC/mL in vascular growth media, put into a sealed top T25 flask, gassed, and left in the incubator overnight on a shaker set to 50 rpm to facilitate parasite egress. The resulting parasite egress efficiency was assessed by a hemocytometer count and blood smear, and was then adjusted to a working concentration of 50 million

ruptured *P. falciparum*-iRBC/mL. The resulting iRBC-egress media was spun at 1000 rpm to remove cellular debris and flash frozen in liquid nitrogen until use. This concentration was chosen as this value is equivalent to a circulating parasitemia of 1%, concentrations frequently found in malaria patients.

### Primary human brain microvascular endothelial cell and primary human brain vascular pericyte culture

Primary HBMEC (ACBRI 376, Cell Systems) were cultured according to the manufacturer's recommendations in complete endothelial growth media-2MV (Lonza) containing 5% fetal bovine serum. Primary HBVP (ScienCell) were cultured according to the manufacturer's recommendations in basal pericyte media supplemented with 1% pericyte growth supplement, 1% penicillin/streptomycin, and 2% fetal bovine serum (ScienCell). Both cell lines were split using Trypsin/EDTA when 90% confluent, seeded on 15 µg/mL poly-L-lysine (P8920, Sigma)-coated T75 flasks, and maintained in a humidified incubator at 37 °C and 5% $CO_2$. Cell lines were tested negative for mycoplasma contamination during the running of experiments. In addition, all experiments were conducted with HBMEC and HBVP with a passage number of 7–9.

### mCherry lentivirus transduction

HBVPs were grown in a T75 flask to a ~90% confluency and incubated with lentiviral particles containing a mCherry vector (kindly donated by Kristina Haase Lab, EMBL Barcelona) in serum-free pericyte media at a multiplicity of infection of 10. After 24 h, the lentivirus particles were removed by washing once every 24-h for a 2-day period. mCherry-positive cells were then selected by fluorescence-activated cell sorting, expanded and froze down for future experiments.

### 3D brain microvessel model fabrication

Type 1 collagen was extracted from rat tails, dissolved in 0.1% acetic acid, lyophilized in a freezer dryer (Labconco Freezone 2.5 Plus), and resuspended at 15 mg/mL in 0.1% acetic acid for storage. The stock collagen-0.1% acetic acid solution was neutralized and diluted to 7.5 mg/mL on ice. A 13 by 13 channel grid or single channel pattern was negatively imprinted into the collagen hydrogel by soft lithography using a PDMS stamp, and inlet and outlets were created by insertion of stainless-steel dowel pins. The top pre-patterned collagen hydrogel contained in a plexiglass top jig was sealed to a flat collagen-layered bottom that sits on a coverslip and a plexiglass bottom jig. The assembly creates perfusable 120 µm diameter microvessels (grid design) or 200-µm diameter microvessels (single channel design), as described previously (Zheng et al, 2012; Piatti et al, 2022). Prior to cell seeding, the channels were incubated with vascular growth media, consisting in EGM-2MV (Lonza) supplemented with 1× pericyte and 1× astrocyte growth factors (ScienCell) for 1 h. Primary HBMECs and HBVPs, or mCherry-positive HBVPs (for pericyte coverage analysis), were resuspended at concentrations of 7 million cells/mL and mixed to a ratio of 3:1 or 5:1 HBMECs to HBVPs. The cell mixture was then seeded into the inlet in 8 µL increments and driven through the microfluidic network by gravity-driven flow. Cells were perfused twice from either the inlet or outlet until the channels were completely covered with adhered cells. Media was then removed and devices were flipped upside down for 1-h to ensure even cell distribution on the top and bottom of the channels.

Microvessels were cultured for 3 days before being used in experiments with media change every twice per day by gravity-driven flow.

### Immunofluorescence microscopy

Microvessel labeling was performed by gravity-driven flow. Microvessels were fixed with 3.7% paraformaldehyde (PFA) in PBS for 15 min followed by three 10-min washes with PBS. Next, microvessels were incubated with Background Buster (Innovex) for 30 min and then permeabilized with blocking buffer (0.1% Triton X-100 and 2% bovine serum albumin in phosphate-buffered saline (PBS)). Primary antibodies including rabbit anti-αSMA-555 (EPR5368, Abcam), mouse anti-vWF (sc-365712, Santa Cruz), mouse anti-CD31 (ab215911, Abcam), mouse anti-ZO-1 (10017242, Invitrogen,), rabbit anti-β-catenin (9587S, Cell Signaling), mouse anti-PDGFRβ (ab69506, Abcam), mouse anti-NG-2 (10424493, Invitrogen), rabbit anti-laminin (ab11575, Abcam), rabbit anti-collagen IV (ab6586, Abcam), mouse anti-VE-cadherin (sc-52751, Santa Cruz), and Phalloidin-647 (A22287, Invitrogen) were diluted in blocking buffer at 1:100 and incubated overnight at 4 °C. After three 10-min PBS washes, secondary antibodies including goat anti-mouse Alexa Fluor 488 (A-11001, Invitrogen), goat anti-rabbit Alexa Fluor 488 (A-11008, Invitrogen), goat anti-mouse Alexa Fluor 594 (A-11005, Invitrogen), goat anti-rabbit Alexa Fluor 594 (A-11012, Invitrogen), goat anti-mouse Alexa Fluor 647 (A-21235, Invitrogen), goat anti-rabbit Alexa Fluor 647 (A-21244, Invitrogen) and 2 mg/mL DAPI (D21490, Invitrogen) were diluted at 1:250 in 2% bovine serum albumin and 5% goat serum containing PBS, and incubated at room temperature for 1 h. Microvessels were washed 6 times with PBS for 10 min each and then imaged on a Zeiss LSM 980 Airyscan 2. Image stacks were acquired with 3 µm or 10 µm z-step size for single or tile scan images, respectively. Z-projections and further threshold analysis was done using Fiji (ImageJ v1.54f) software. For 2D imaging, 50000 HBMECs or HBVPs were seeded into µ-Slide 8-well high glass bottom chambered coverslips (80807, IBIDI) and grown for 3 days, fixed, and labeled as described above.

### Pericyte coverage analysis

Tile scan images of microvessel networks with mCherry-positive HBVP and vWF-labeled HBMEC were acquired as described above. Images were divided into two stacks, one being the top surface of the microvessels of the network and the other the bottom surface. Next, Z-projections of the top and bottom surfaces of the network were generated. The percentage of endothelial surface that was covered by pericytes was calculated by creating a mask of both the mCherry HBVP and the vWF-positive HBMEC.

### Serial block-face scanning electron microscopy

Microvessels were grown for 3 days, then treated with iRBC-egress media or vascular growth media for 18 h, and then fixed by adding 2% PFA and 2.5% glutaraldehyde (GA) in EGM-2MV to the device inlet for 30 min and washed twice with EGM-2MV at 37 °C. The collagen hydrogel was then carefully removed from the plexiglass jig and microvessel regions exposed to low wall shear stress were cut out and fixed with a secondary fixative solution (2% PFA, 2.5% GA, 0.25 mM $CaCl_2$, 0.5 mM $MgCl$, 5% sucrose in a pH 7.4 0.1 M Cacodylate buffer) overnight at 4 °C, and then rinsed twice for 15-min with 0.1 M Cacodylate buffer. Samples were post-fixed in a

reduced Osmium solution (1% $OsO_4$, 1.5% $K_3FeCN_6$ in 0.065 M Cacodylate buffer) for 2 h at 4 °C followed by six 10-min washes in $dH_2O$. Post-staining consisted of subsequent incubation steps of 1% thiocarbohydrazide in $dH_2O$, 2% Osmium tetroxide in $dH_2O$, and 1% Uranyl acetate in $dH_2O$, aided by a PELCO Biowave Pro+ (Ted Pella) containing a SteadyTemp Pro and ColdSpot set to 20 °C at $7 \times 2$ min cycling on-off at 100 W under vacuum, with $dH_2O$ rinses in between steps once in a fume hood and twice in the microwave at 250 W for 40 s without vacuum. The stained samples were then dehydrated by serial additions of 30, 50, 80, and 100% ethanol solutions in the microwave at 250 W for 40 s without vacuum with ColdSpot set to 4 °C, infiltrated with EPON 812 hard epoxy resin in steps of 25, 50, 75, 90, and $3 \times 100\%$ resin diluted in ethanol with each step in the microwave at 150 W for 3 min under vacuum and polymerized at 60 °C for 48 h. The sample was trimmed (UC7, Leica Microsystems) using a 90° cryo-trimmer (Diatome) to generate a small block face. The resulting resin block was mounted on a pin stub using silver conductive epoxy resin (Ted Pella). The SBF-SEM acquisition was performed with a Zeiss Gemini2 equipped with a Gatan 3view microtome and a focal charge compensation device (Zeiss). The SEM was operated at 1.5 kV 300 pA, using a pixel size of 15 nm, a slice thickness of 50 nm and a dwell time of 1.6 μs. The SBF-SEM acquisition was performed using the software SBEMimage (Titze et al, 2018). The acellular center of the lumen was not imaged to save imaging time. Following acquisition, the positional metadata of the sections and tiles obtained was converted with SBEM-specific import scripts (https://git.embl.de/schorb/rendermodules-addons) for subsequent assembly, stitching and alignment of large EM datasets (Mahalingam et al, 2022). Aligned images were then binned to 22 by 22 by $250 \text{ nm}^3$ in xyz and segmentation was then performed using the software package Amira (Thermo Fischer Scientific). Primary HBVP and HBMEC were segmented manually using the magic wand and brush tool to segment every fifth section, followed by the use of the interpolation tool to segment the entire volume. The resulting segmented microvessels were rendered using the generate surfaces module for 3D visualization.

### Serial block-face scanning electron microscopy pericyte morphology analysis

3D-rendered pericyte meshes, segmented from SBF-SEM images of microvessels treated with either media alone or iRBC-egress media, were generated using Amira. Following segmentation, the meshes were exported as *STL* files and processed with a custom-developed Python script, designed to quantify aspects of pericyte morphology, including mesh circularity and maximum width (https://github.com/waleedmirzaPhD/cell_analysis_toolkit). First, the 3D pericyte meshes were mapped onto a 2D plane through principal component analysis (PCA), effectively reducing the data's dimensionality while retaining its most significant morphological features. This mapping uses the two principal components that encapsulate the dimensions with the maximum variance in the 3D mesh, thus ensuring the conservation of structural information. Subsequently, convex hull analysis (Virtanen et al, 2020) was applied to delineate the pericyte mesh boundaries, providing a clear representation of each structure's shape by which circularity and maximum width could be measured. The circularity of each mesh is calculated using the convex hull, by assessing the ratio of the area enclosed by the hull to the square of the hull perimeter, with the formula for

circularity given as $4\pi \times Area/Perimeter^2$. This circularity metric provides insight into the roundness of the mesh, where a value closer to 1 indicates a shape that is more circular, and values closer to 0 reflect more elongated or irregular shapes. Additionally, the maximum width is defined by identifying the two farthest points along the convex hull boundary and then calculating the greatest perpendicular distance from this line (connecting the farthest points) to any point within the convex hull. This measurement represents the maximum width of the pericyte mesh by assuming its length is the axis with the greater size. In addition, the number of pericyte branches was calculated manually by counting the number of branches on each 3D-rendered pericyte. Pericytic branches were defined as cellular protrusions greater than 10 μm in length that bifurcate from the main cell body, as described previously (Hartmann et al, 2015).

### P. falciparum-iRBC and iRBC-egress media perfusion to 3D brain microvessels

All experiments were done in 3D microvessel devices grown for 3 days. Schizont-stage *P. falciparum*-iRBC were purified to >60% parasitemia by 40 mg/mL gelaspan, and the resulting enriched population was diluted in vascular growth media to 50 million iRBC/mL (same concentration as the *P. falciparum*-iRBC-egress media). In all, 200 μL of *P. falciparum*-iRBC or *P. falciparum*-iRBC-egress media was added into the device inlet and perfused by gravity flow for 30 min with the outlet effluent being reintroduced to the inlet every 10 min. Microvessels treated with intact *P. falciparum*-iRBC (but not *P. falciparum*-iRBC-egress media), were then washed for 10 min with vascular growth media. Both conditions were incubated overnight for 18 h. Supernatant was then removed from the outlet and pooled for Luminex analysis, and 3D brain microvessels were used for permeability studies or fixed for imaging with 3.7% PFA for 15 min (see section "Immunofluorescence microscopy").

### Fluorescent dextran-based permeability assay

Single-channel 5:1 HBMEC:HBVP or HBMEC-only microvessel devices were grown for 3 days, then perfused and incubated for 18 h with either vascular growth media or iRBC-egress media as described above. To determine resulting changes in permeability, they were perfused with a 70 kDa FITC-dextran solution as follows: the device was washed 1× with PBS and placed in a Zeiss LSM 980 Airyscan2 confocal microscope with a temperature and $CO_2$-controlled imaging chamber (37 °C, 5% $CO_2$). PBS was aspirated, and 100 μL of 70 kDa FITC-dextran solution (100 μg/mL) in PBS was pulled from the inlet to the outlet at 6.5 μL/min flow rate using a syringe pump (Harvard Apparatus PHD 2000). Tilescan confocal images of the entire channel were taken every 30 s for 5 min once the channel had been filled with FITC dextran. In devices pre-treated with 1 μg/mL recombinant Ang-1 (923-AB-025, R&D Systems), pre-incubation occurred for 18 h prior to iRBC-egress media or media addition, starting on day 2. rAng-1 permeability experiments were done in the absence of fetal bovine serum, as this is an additional Ang-1 source present in media (Fig. EV4C). In devices pre-treated with 10 μM AKB-9778 (HY-1009041-1MG, MedChemExpress), pre-incubation occurred 1 h prior to iRBC-egress media or media addition. To calculate the percentage of recovery with either rAng-1 or AKB-9778, we divided the median permeability values of rAng-1/AKB-9778 pre-treated microvessels in the presence of iRBC-egress media by the permeability values of

control rAng-1/AKB-9778 pre-treated microvessels in the absence of iRBC products. These values were compared with the same normalization done in untreated devices with and without iRBC-egress media.

### Quantification of apparent permeability

Apparent permeability is determined as the flux of fluorescently labeled dextran across the microvessel wall into the surrounding collagen hydrogel. All steps of permeability analysis were performed using ImageJ. By using the following equation:

$$Papp = \frac{1}{I_{Vt1} - I_{Tt1}} \times \frac{I_{Tt2} - I_{Tt1}}{\Delta t} \times \frac{A_{lateral\,tissue}}{P}$$

The apparent permeability ($P_{app}$) was calculated using the fluorescence intensity inside the vessel ($I_V$) and in the surrounding collagen hydrogel ($I_T$) at two time points t1 (0 min after the vessel is filled with dextran) and t2 (5 min after the vessel is filled). $\Delta t$ is the change in time, $A_{lateral\,tissue}$ is the area of the collagen hydrogel being examined and P is the perimeter of the microvessel. To account for *P. falciparum*-iRBC-egress material that appears dark even in the presence of 70 kDa FITC-dextran, regions of *P. falciparum*-iRBC-egress material were excluded from the calculation of the fluorescence intensity inside the vessel ($I_V$) by using ImageJ median smoothing and the magic wand tool to select only the area of the channel without *P. falciparum*-iRBC-egress material. Apparent permeability was analyzed from both microvessel walls of the channel, yielding two values for every channel.

### Western blot

In total, $10^5$ HBMECs were seeded into a six-well plate and allowed to grow for 3 days. On day 3, the cells were incubated for 18 h with either media-only or iRBC-egress media. Cells were then washed once with 4 °C PBS and lysed with 4 °C Pierce RIPA buffer (89900, Invitrogen) and 1× Protease inhibitor (A32963, Invitrogen) for 15 min. The cellular components were lifted into suspension using a cell scrapper, and the resulting cell lysates were spun at $18,000 \times g$, 4 °C to remove cellular debris. Protein concentrations were quantified using a BCA assay (23227, Invitrogen) following the manufacturer's protocols, and 20 mg of protein was ran on precast Mini-PROTEAN TGX gels (4561034, Bio-Rad), along with 5 μL of Odyssey One-Color Protein Molecular Weight Marker (928-40000, LI-COR Biotech) or Precision Plus Protein WesternC Blotting Standard (1610376, Bio-Rad) at 170 V for 1 h. Gels were transferred to nitrocellulose membranes (1704159, Bio-Rad) using a Trans-blot Turbo transfer machine set to mixed molecular weight (2.5 A, 25 V). Membranes were blocked in 5% BSA-TBS for 2 h on a shaker set to 50 rpm and then incubated with 1:1000 rabbit anti-claudin-5 (E8F3D, Cell signaling), 1:1000 rabbit anti-VE-cadherin (D87F2, Cell signaling) or 1:1000 mouse anti-β-tubulin (MA5-16308, Invitrogen) in 3% BSA-Tris-buffered saline (TBS) + 0.2% Tween-20 at 4 °C, overnight on a shaker. After three 5 min TBS + 0.2% Tween-20 washes, membranes were incubated with 1:10,000 IRDye goat anti-mouse 680RD (926-68070, LI-COR Biotech) and 1:20,000 IRDye goat anti-rabbit 800CW (926-32211, LI-COR Biotech) in 3% BSA-TBS + 0.2% Tween-20 + 0.01% sodium dodecyl sulphate for 1 h at RT on a shaker. Membranes were then washed 4 times with TBS + 0.2% Tween-20 for 5 min each, rinsed in $H_2O$, and imaged using a LI-COR Odyssey M scanner. Densitometry analysis was performed on acquired images using Fiji where a ROI was manually defined to encircle each band at the correct molecular weight (23 kDa for claudin-5, 120–140 kDa for VE-cadherin, and 55 kDa for β-tubulin), including an ROI in an area of the membrane with no bands for background signal subtraction. Background integrated density was subtracted from each protein band and the resulting target protein integrated density value was normalized to that of the corresponding β-tubulin loading control using division.

### Co-culture transwell fabrication

In all, $10^5$ HBVPs were seeded on the basolateral side of poly-L-lysine-treated transwell inserts (12-well PET membrane inserts with a pore size of 3 μm) in 100 μL of media and left to attach for 4–6 h. Transwell inserts were then flipped into a 12-well culture plate containing 1200 μL of pericyte media and allowed to grow for 2 days. Next, $10^5$ HBMECs were seeded onto the apical side of the poly-L-lysine-treated transwell inserts in 300 μL of media and left to attach for 4–6 h. Then media was removed from both sides of the insert, and 1200 μL of pericyte media was added to the basolateral side and 800 μL of EGM-2MV media to the apical side. The cells were allowed to grow for 2 more days. On day 5, supernatants were taken from the basolateral side for angiopoietin-1 ELISA or the apical side for angiopoietin-2 ELISA. Endothelial cell-only transwells underwent the same fabrication protocol minus the initial pericyte addition steps.

### Angiopoietin-1 and -2 ELISA

Ang-1 and Ang-2 protein concentrations were measured from supernatants taken from the basolateral or apical side of the transwell model, respectively. Supernatant concentrations were obtained by running 5× diluted samples on either the Human Angiopoietin-1 DuoSet ELISA kit (DY923, R&D Systems) or the Human Angiopoietin-2 DuoSet ELISA kit (DY623, R&D Systems). Protein concentrations were interpolated from sigmoidal 4-parameter-fit standard curves generated from a 7-point standard curve of recombinant human proteins using GraphPad Prism.

### Quantification of secreted proteins by Luminex Assay

Secreted protein concentrations were measured from supernatants taken from the outlets of 3D brain microvessels with a $13 \times 13$ grid geometry. Supernatants from two to three devices exposed to the same condition were pooled, diluted 2× and assayed using a 10-plex Human Luminex Discovery Assay from R&D Systems on a Luminex 100/200. The 10-plex panel included: Ang-1, Ang-2, PDGF-BB, N-cadherin, TIMP-1, ANGPTL4, VEGF, Tie-2, IL-8, and CXCL-1. Protein concentrations were interpolated from a 11-point standard curve of known concentrations of recombinant human proteins provided by the vendor and reported as pg/mL using xPONENT 4.2. 7 out of 10 analytes were in the standard curve quantification range for 100% of measurements recorded, with Timp-1, N-cadherin, and IL-8 having 100% of measurements above the upper limit of quantification.

### Measuring changes in barrier integrity by xCELLigence

96-well PET E-plates (300600910, Agilent) were coated with 15 μg/mL poly-L-lysine, and 5000 HBMECs in 200 μL EGM-2MV were added per well. After cell adherence, the plate was placed into the xCELLigence RTCA SP reader to begin baseline measurement of growth-related changes in cell index (an arbitrary measure of impedance), and the cells were incubated for 3 days until cell index reached a plateau (indicative of cell confluence). Media was changed every two days. For initial testing of the impact of iRBC-egress media on barrier integrity, on the day of the

### The paper explained

#### Problem

Cerebral malaria (CM) is a severe complication of *Plasmodium falciparum* infection, resulting in the majority of ~600,000 malaria annual deaths. Despite anti-malarial drug administration upon hospitalization, fatality rates still range from 15 to 25% and many survivors suffer long-term neurological disabilities. A common dysregulated vascular pathway identified in CM patients is the angiopoietin–Tie axis. Treatments that restore this vascular homeostatic pathway appear as a potential avenue for adjunctive therapies in experimental rodent CM models. Nevertheless, the use of rodent CM models for therapeutic discovery is not ideal, given that *P. falciparum* pathogenesis is species-specific. Therefore, the development of novel and advanced human 3D microvascular models offers new avenues to study disease pathogenesis and explore potential adjunctive CM treatments.

#### Results

In this study, we generate a 3D human brain microvasculature model that reproduces in vivo interactions between two key cell types necessary to reproduce the protective angiopoietin–Tie axis: human brain endothelial cells and pericytes. The addition of *P. falciparum*-infected red blood cell (iRBC) egress products causes vascular disruption and hampers the release of the vascular protective factor, angiopoietin-1 (Ang-1), from brain pericytes. Rescue with recombinant Ang-1 partially prevents iRBC egress product-induced vascular disruption. A short 1-h pre-treatment of the microvessels with AKB-9778, a downstream pharmaceutical inducer of angiopoietin–Tie axis activity currently in phase II clinical trial for diabetic retinopathy, significantly restores vascular integrity. Our study highlights the role of pericytes in CM and the therapeutic potential of interventions that restore the angiopoietin–Tie axis as adjunctive therapeutics.

#### Impact

Our study demonstrates the potential of bioengineered vascular models to recapitulate dysregulated pathways previously characterized in malaria patients, and in providing a physiologically relevant platform to test adjunctive therapies. The use of the 3D brain microvascular model has enhanced our understanding of the mechanisms behind CM pathogenesis, uncovering a previously unappreciated effect of *P. falciparum* on brain pericytes, linking angiopoietin–Tie axis dysregulation and microvasculature disruption. These findings pave the way for the identification of novel, fast-acting therapeutics, such as AKB-9778, to restore vascular integrity in CM patients.

experiment, iRBC-egress media was added at concentrations of either 12.5, 25, or 50 million ruptured iRBC/mL. Changes in cell index were measured every 15 s for 8 h and then every 15 min for 48 h. To test the impact of rAng-1 or AKB-9778 on barrier breakdown mediated by *P. falciparum*-iRBC-egress, iRBC-egress media was added at $50 \times 10^6$ ruptured iRBC/mL after either a 1- or 18-h pre-incubation with 1 µg/mL of rAng-1 or 10 µM of AKB-9778. Again, changes in cell index were measured every 15 s for 8 h and then every 15 min for 48 h. Quantification of partial protection was done by first normalizing each condition to the media-only control. Then the area under the curve of the negative values, indicative of barrier breakdown, was calculated using GraphPad Prism (version 10.0.2), where the media-only without rAng-1 condition represented an average value of 0 and the iRBC-egress media without rAng-1 condition an average value of 100.

### Statistical analysis

All statistics were obtained using GraphPad Prism (version 10.3.0). Nonparametric Mann–Whitney *U* tests were performed for most experiments, except to calculate Ang-1 protection in the 2D xCELLigence and microvessel experiments, in which a repeated measures one-way ANOVA test with Dunnett's multiple comparisons test and Kruskal–Wallis test with Dunn's multiple comparisons test were performed, respectively. A *P* value < 0.05 was considered statistically significant.

## Data availability

The SBF-SEM dataset produced in this study is available in the following database record:SBF-SEM dataset: EMPIAR dataset under accession code EMPIAR-12967. The source data of this paper are collected in the following database record: https://www.ebi.ac.uk/biostudies/bioimages/studies/S-BIAD2217?query=S-BIAD2217.

The source data of this paper are collected in the following database record: biostudies:S-SCDT-10_1038-S44321-025-00319-y.

## Peer review information

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

## Acknowledgements

We want to thank Kristina Haase (EMBL Barcelona) for her support and careful suggestions on the project and manuscript writing, as well as for assistance with the generation of mCherry HBVP with help from Violeta Beltran-Sastre. We acknowledge access to the CRG/UPF flow cytometry facility for mCherry HBVP sorting. We are grateful to Carlota Dobaño (ISGlobal) for access to the Luminex and to Michael Blackman (The Francis Crick Institute), who kindly gifted compound 2. This work was facilitated by the EMBL Electron Microscopy Core Facility (EMCF), with important assistance from Viola Oorschot on sample preparation and processing, and Karel Mocaer and Martin Schorb on image analysis. The great majority of this work was supported by the core program funding of the European Molecular Biology Laboratory (EMBL) and the European Research Council (ERC) under the European Union's Horizon 2020 research and innovation program (Grant agreement no. 948088). FK is funded through the Marie Skłodowska-Curie grant agreement (101068552). HF is supported by a fellowship from the EMBL Interdisciplinary (EI4POD) program under Marie Skłodowska-Curie Actions COFUND (847543). GM is supported by RYC 2020–029886 I/AEI/10.13039/501100011033, co-funded by European Social Fund (ESF). ISGlobal received support from the grant CEX2018-000806-S funded by MCIN/AEI/ 10.13039/501100011033, and support from the Generalitat de Catalunya through the CERCA Program. This research is part of the ISGlobal's Program on the Molecular Mechanisms of Malaria which is partially supported by the Fundación Ramón Areces.

## Author contributions

**Rory K M Long**: Conceptualization; Data curation; Formal analysis; Investigation; Visualization; Methodology; Writing—original draft; Writing—review and editing. **François Korbmacher**: Investigation; Methodology; Writing—original draft; Writing—review and editing. **Paolo Ronchi**: Resources; Investigation; Methodology; Writing—original draft; Writing—review and editing. **Hannah Fleckenstein**: Resources; Investigation; Methodology; Writing—original draft; Writing—review and editing. **Martin Schorb**: Software; Writing—original draft; Writing—review and editing. **Waleed Mirza**: Software; Writing—original draft; Writing—review and editing. **Mireia Mallorquí**: Investigation; Writing—original draft; Writing—review and editing. **Ruth Aguilar**: Resources; Investigation; Methodology; Writing—original draft; Writing—review and editing. **Gemma Moncunill**: Resources; Writing—original draft; Writing—review and editing. **Yannick Schwab**: Resources; Methodology; Writing—original draft; Writing—review and editing. **Maria Bernabeu**: Conceptualization; Resources; Supervision; Funding acquisition; Writing—original draft; Project administration; Writing—review and editing.

Source data underlying figure panels in this paper may have individual authorship assigned. Where available, figure panel/source data authorship is listed in the following database record: biostudies:S-SCDT-10_1038-S44321-025-00319-y.

## Funding

## Disclosure and competing interests statement

The authors declare no competing interests.

# Expanded View Figures

**Figure EV1. Characterization of brain-specific endothelial and pericyte marker expression and secretion of angiopoietin–Tie axis components.**

(A) Immunofluorescence maximum z-projection of a 2D HBMEC monolayer stained for adherens and tight junctional markers: β-catenin, VE-cadherin, ZO-1 (top) and claudin-5 (bottom), and 4′, 6-diamidino-2-phenylindole (DAPI). The merged image includes the adherens junction markers (white), tight junction markers (red) and DAPI labeling (blue). Scale bars: 50 μm. (B) Immunofluorescence maximum z-projection of a 2D HBMEC monolayer stained for vWF, brain glucose transporter GLUT-1, CD31, VE-cadherin and DAPI. The merge image includes vWF or GLUT-1 (white), CD31 or VE-cadherin (red) and DAPI labeling (blue). Scale bars: 50 μm. (C) Immunofluorescence maximum z-projection of a 2D HBVP monolayer stained for PDGFRβ (top) and NG-2 (bottom), Phalloidin and DAPI. The merge staining includes the pericyte markers (white), phalloidin (red) and DAPI labeling (blue). Scale bars: 50 μm. (D) Concentration of secreted angiopoietin-1 (top) and angiopoietin-2 (bottom) in supernatant obtained from either HBVP-HBMEC co-culture or HBMEC-only monolayers grown in a transwell model. Data is presented as mean $+/-$ standard deviation ($n = 3$ independent experiments for angiopoetin-1 measurement and 4 for angiopoietin-2 measurement, Mann–Whitney $U$ test).

▶

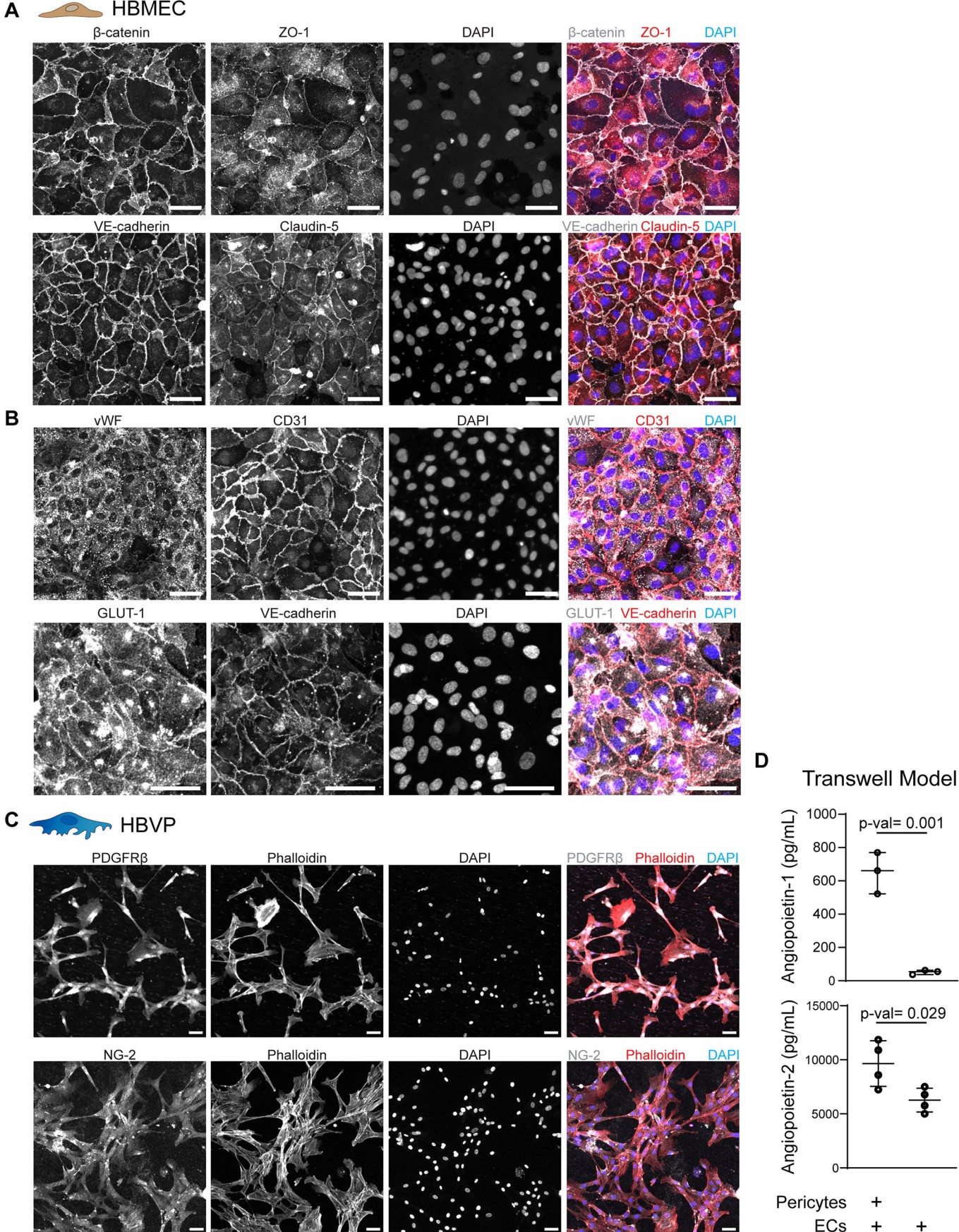

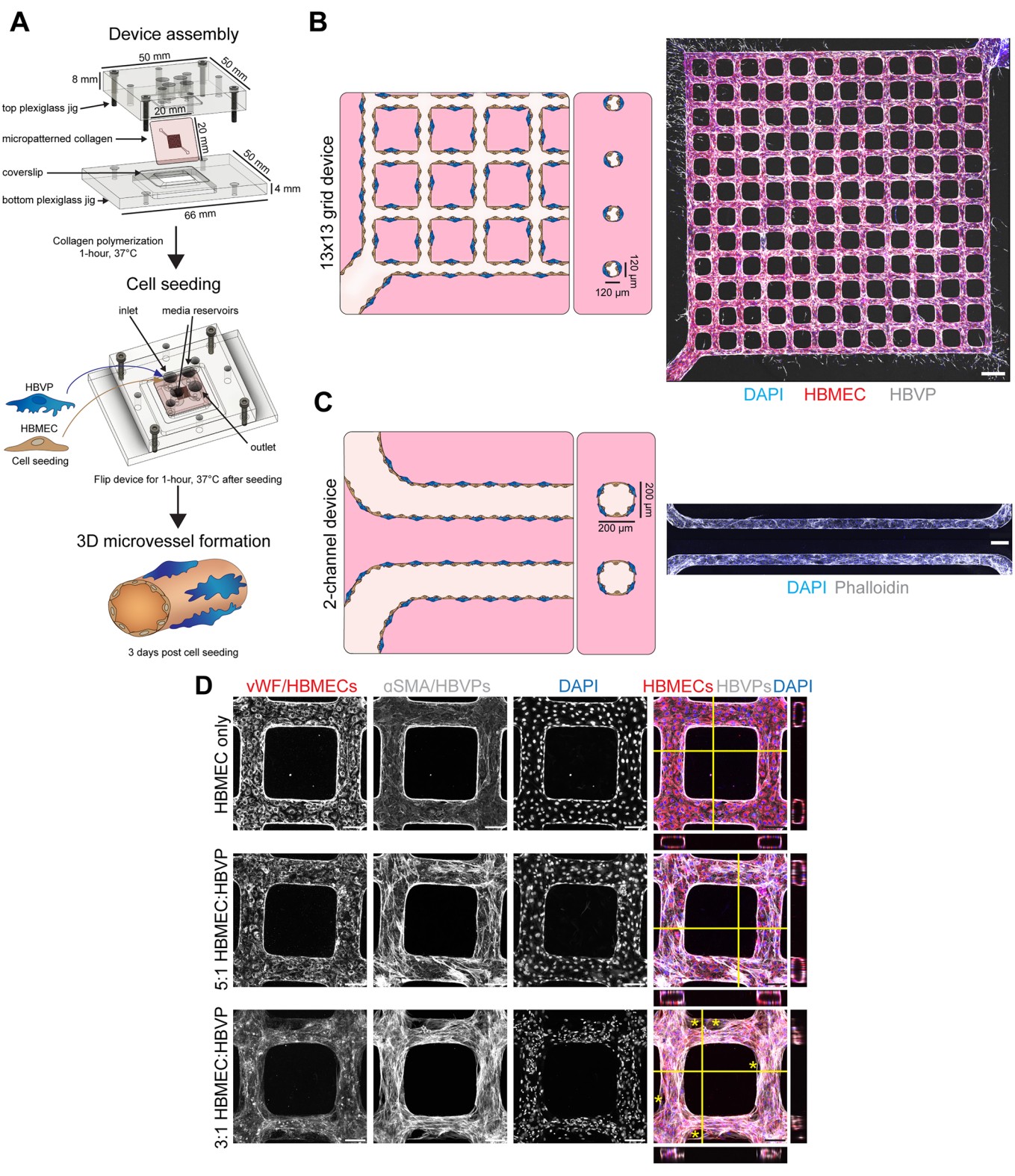

◀  **Figure EV2.  Fabrication of the 3D brain microvessel model.**

(**A**) Schematic representation of the protocol to fabricate the 3D brain microvessel model including device assembly, cell seeding and 3D microvessel formation.
(**B**) Schematic overview and cross section of the 13 × 13 grid microvessel network (left). IFA maximum z-projection of the complete grid microvessel network stained for vWF for endothelial cells (red), mCherry for pericytes (white) and DAPI (blue). Scale bar: 500 µm. (**C**) Schematic overview and cross section of the 2-channel microvessel network (left). IFA maximum z-projection of the complete grid microvessel network stained for phalloidin (white) and DAPI (blue). Scale bar: 200 µm.
(**D**) Immunofluorescence assay (IFA) maximum z-projection of microvessels fabricated using HBMEC-only, a 3:1 or 5:1 HBMEC to HBVP ratio labeled with vWF for HBMEC (red), αSMA for HBVP (white) and DAPI (blue). Orthogonal views display the presence or absence of a perfusable microvessel lumen and yellow asterisks represent regions of microvessel detachment. Scale bar: 100 µm.

                                                                                               

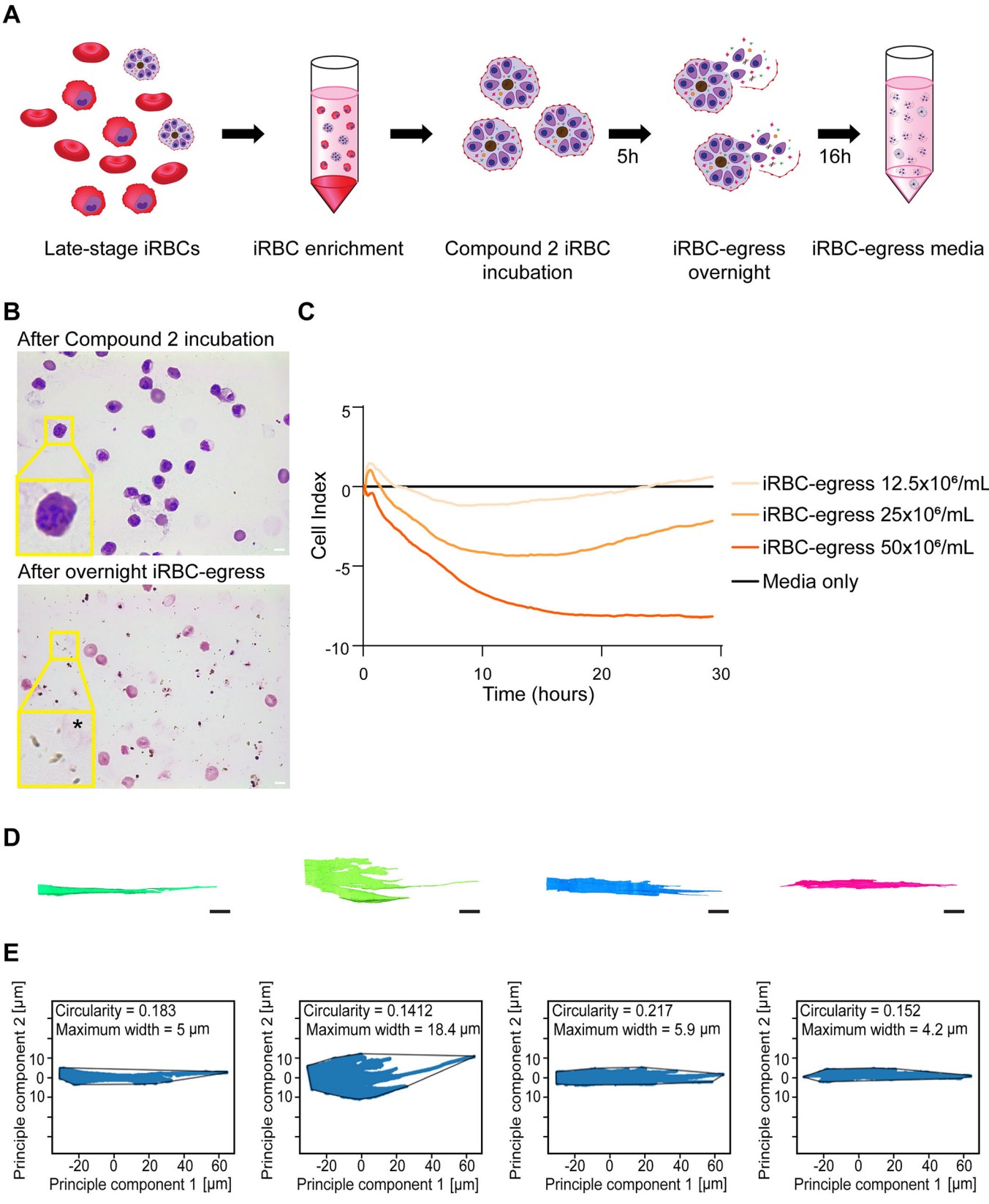

**A**

Late-stage iRBCs → iRBC enrichment → Compound 2 iRBC incubation →(5h) iRBC-egress overnight →(16h) iRBC-egress media

**B**

After Compound 2 incubation

After overnight iRBC-egress

**C**

iRBC-egress 12.5×10⁶/mL
iRBC-egress 25×10⁶/mL
iRBC-egress 50×10⁶/mL
Media only

**D**

**E**

Circularity = 0.183
Maximum width = 5 µm

Circularity = 0.1412
Maximum width = 18.4 µm

Circularity = 0.217
Maximum width = 5.9 µm

Circularity = 0.152
Maximum width = 4.2 µm

◀ **Figure EV3. Generation of endothelial barrier disruptive iRBC-egress media and a pipeline for analysis of pericyte morphological features.**

(A) Schematic representation of the protocol to make iRBC-egress media. In short, late-stage *P. falciparum*-iRBC are purified by a gelaspan gradient separation and then incubated for 5 h with compound-2 to synchronize them at the point of egress. Compound-2 is removed and the *P. falciparum*-iRBC are resuspended in vascular growth media and left overnight on a shaker at 50 rpm to egress. (B) A thin smear of tightly synchronized schizonts before the removal of Compound-2 (yellow inset highlights a ROI with a schizont-stage iRBC) or the resultant iRBC-egress media before centrifugation (yellow inset highlights a ROI with free hemozoin particles next to an iRBC ghost, denoted with an asterisk) stained with Giemsa. Scale bars: 5 μm. (C) Representative recording data of xCELLigence measurements on 2D HBMEC monolayers in the absence of HBVP taken after addition of iRBC-egress media at different concentrations. Data is normalized to the media-only control. (D) The analysis pipeline begins with the extraction of 3D segmented pericyte meshes. Shown here are four representative pericytes. Scale bars: 10 μm. (E) Pericyte geometry is flattened into two principal dimensions using PCA by a Python-based image analysis pipeline. The 2D geometry's shape is then determined using convex hull analysis. To describe pericyte morphology, pericyte cell borders are analyzed for circularity, calculated as $4\pi*Area/Perimeter^2$, where a value of 1 indicates a perfect circle and values approaching 0 indicate increasingly elongated shapes, and maximum width.

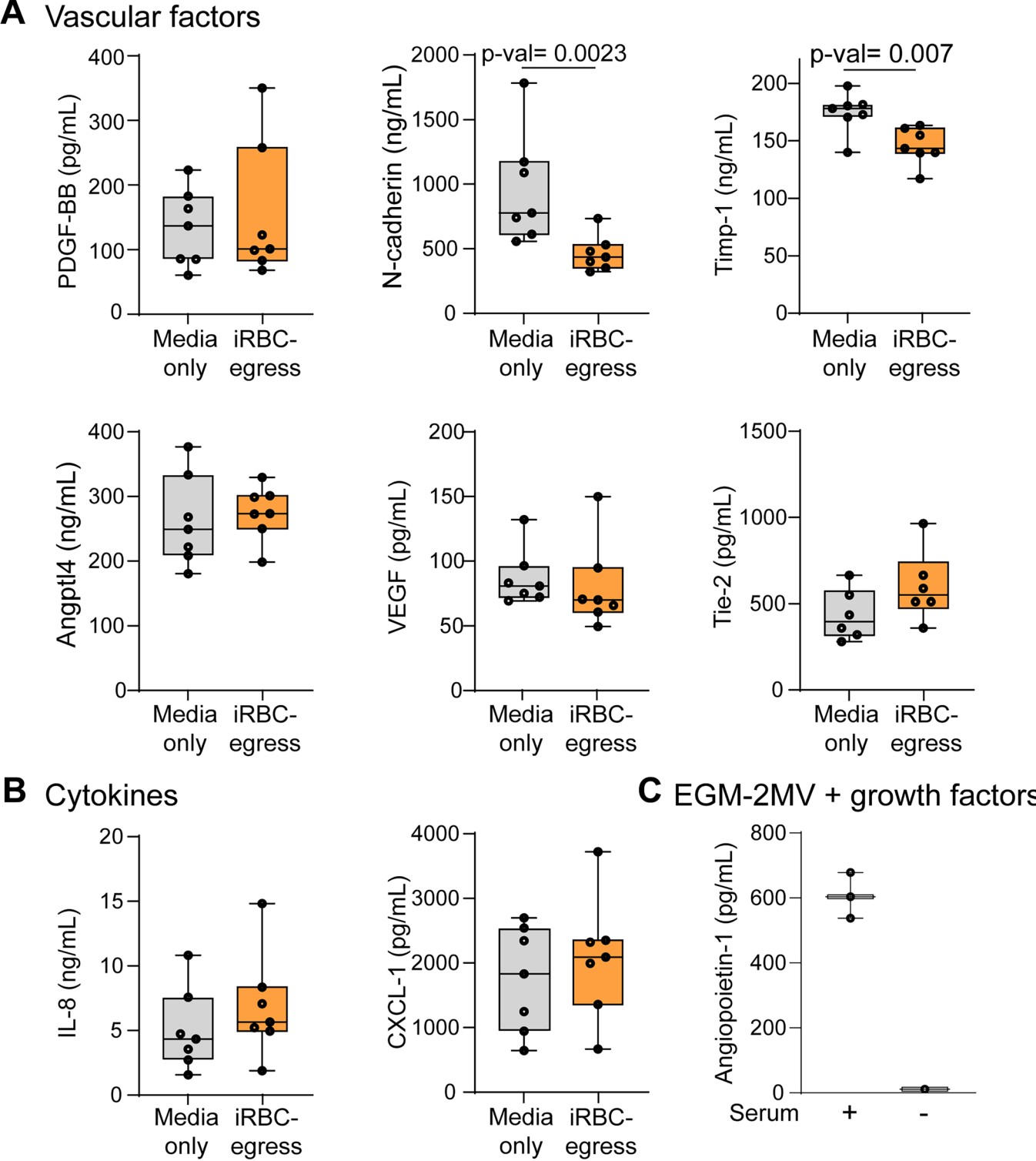

**Figure EV4. iRBC egress products cause alteration of endothelial cell-pericyte interaction markers.**

(A) Concentrations of vascular factors PDGF-BB, N-cadherin, Timp-1, Angptl4, VEGF and Tie-2 measured by Luminex from 3D brain microvessels supernatants treated with media-only or iRBC-egress media for 18-h. Box and whisker plots display the median, 25th and 75th percentiles and the minimum and maximum data points. ($n = 7$ supernatants pooled from 2–3 devices each). (B) Concentrations of released cytokines IL-8 and CXCL-1 measured by Luminex from 3D brain microvessels supernatants treated with media-only or iRBC-egress media for 18-h. Box and whisker plots display the median, 25th and 75th percentiles and the minimum and maximum data points. ($n = 7$ supernatants pooled from 2-3 devices each). (C) Concentrations of angiopoietin-1 measured by Luminex from freshly made growth factor-containing EGM-2MV with or without serum. Box and whisker plots display the median, 25th and 75th percentiles and the minimum and maximum data points. ($n = 3$, where angiopoietin-1 in serum-free vascular media was below the limit of detection). Data information: Statistical significance is analyzed by Mann–Whitney $U$ test (A, B).

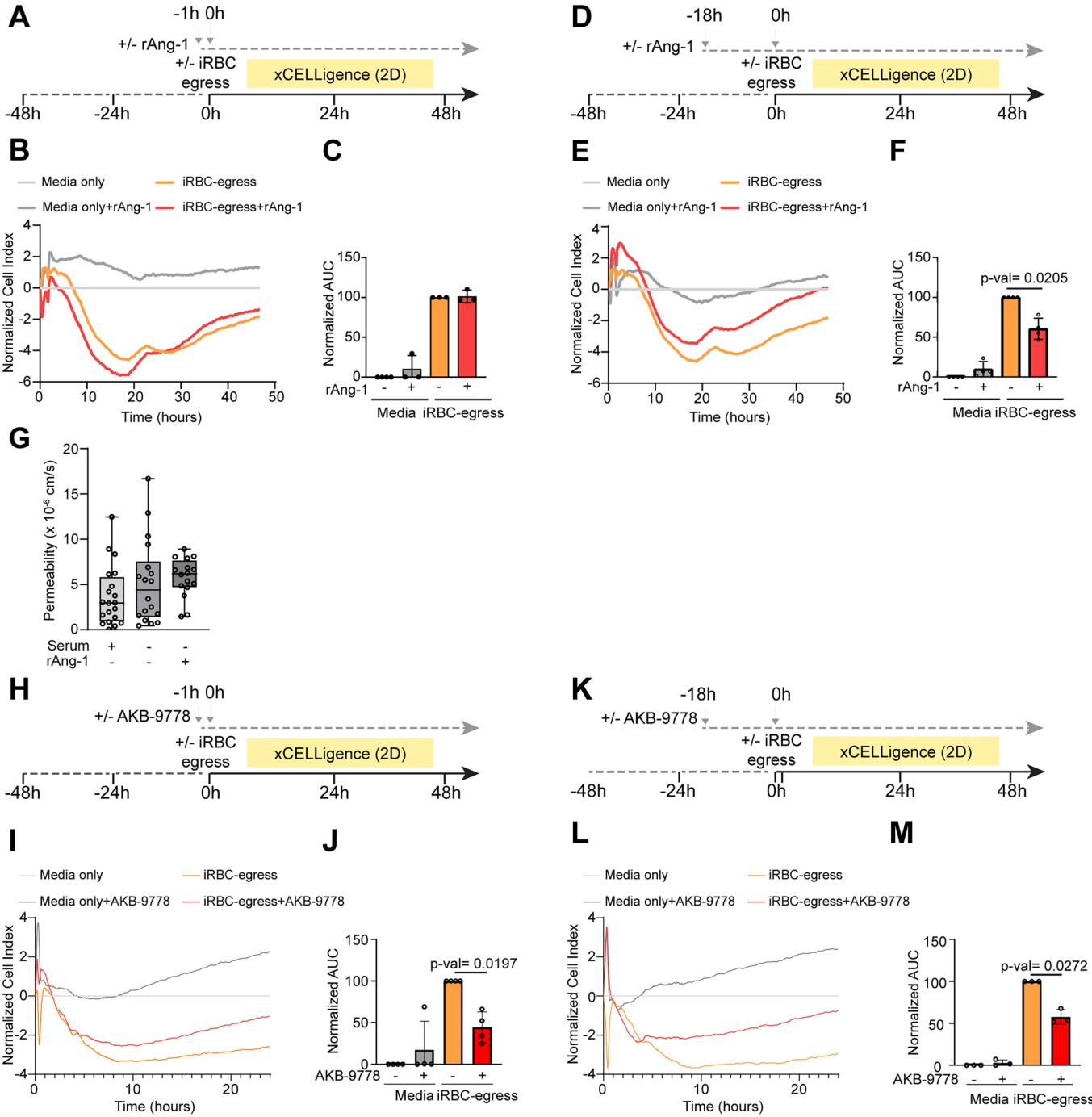

◄ **Figure EV5. Recombinant Ang-1 or AKB-9778 pre-treatment partially protects against 2D endothelial monolayer permeability increase induced by iRBC egress products.**

(A) Experimental outline of short-term (1-h) incubation with rAng-1 in a 2D HBMEC monolayer in the absence of HBVP. (B) Representative recording data of xCELLigence measurements taken after a 1-h +/-rAng-1 pre-treatment followed by +/- iRBC-egress media addition. All conditions were normalized to the media-only control. (C) Area under the curve analysis with iRBC-egress media normalized as 100 ($n = 3$ independent experiments run in triplicate). (D) Experimental outline of long-term (18-h) incubation with rAng-1 in a 2D HBMEC monolayer in the absence of HBVP. (E) Representative recording data of xCELLigence measurements taken after an 18-h +/− rAng-1 pre-treatment followed by +/− iRBC-egress media addition. All conditions were normalized to the media-only control. (F) Area under the curve analysis with iRBC-egress media normalized as 100. ($n = 4$ independent experiments run in triplicate). (G) Apparent permeability of 70 kDa FITC-dextran in microvessels pre-treated with serum containing media, serum-free media or serum-free media+rAng-1 ($n = 10$ individual devices for serum containing media incubation, 9 for serum-free media incubation and 8 for serum-free media+rAng-1 incubation). (H) Experimental outline of short-term (1-h) incubation with AKB-9778 in a 2D HBMEC monolayer in the absence of HBVP. (I) Representative recording data of xCELLigence measurements taken after a 1-h +/−AKB-9778 pre-treatment followed by +/− iRBC-egress media addition. All conditions were normalized to the media-only control. (J) Area under the curve analysis with iRBC-egress media normalized as 100 ($n = $ of 4 independent experiments run in triplicate). (K) Experimental outline of long-term (18-h) incubation with AKB-9778 in a 2D HBMEC monolayer in the absence of HBVP.
(L) Representative recording data of xCELLigence measurements taken after an 18-h +/− rAng-1 pre-treatment followed by +/− iRBC-egress media addition. All conditions were normalized to the media-only control. (M) Area under the curve analysis with iRBC-egress media normalized as 100. ($n = 3$ independent experiments run in triplicate). Data information: In (C, F, J, M), data are presented as bar charts displaying the mean +/- standard deviation (Repeated measures one-way ANOVA test with Dunnett's multiple comparisons test. In (G), box and whisker plots display the median, 25th and 75th percentiles and the minimum and maximum data points (Kruskal–Wallis test corrected for multiple comparisons using the Benjamini, Krieger and Yekutieli method).

