## [Peer Review File · EMBO Molecular Medicine]

Plasmodium falciparum impairs Ang-1 secretion by pericytes in a 3D brain microvessel model

Rory Long, François Korbmayer, Paolo Ronchi, Hannah Fleckenstein, Martin Schorb, Waleed Mirza, Mireia Mallorquí, Ruth Aguilar, Gemma Moncunill, Yannick Schwab, and Maria Bernabeu Aznar

Corresponding author: Maria Bernabeu Aznar (maria.bernabeu@embl.es)

Review Timeline:

Submission Date:	7th Feb 25
Editorial Decision:	3rd Mar 25
Revision Received:	1st Aug 25
Editorial Decision:	4th Sep 25
Revision Received:	24th Sep 25
Accepted:	29th Sep 25

Editor: Zeljko Durdevic

Transaction Report:

3rd Mar 2025

Dear Dr. Bernabeu Aznar,

Thank you for the submission of your manuscript to EMBO Molecular Medicine. We have now received feedback from the three reviewers who agreed to evaluate your manuscript. All three referees recognize interest of the study but also raise important concerns that should be addressed in a major revision. If you would like to discuss further the points raised by the referees, I am available to do so via email or video. Let me know if you are interested in this option.

We would welcome the submission of a revised version within three months for further consideration. Please let us know if you require longer to complete the revision.

Please use this link to login to the manuscript system and submit your revision: <https://embomolmed.msubmit.net/cgi-bin/main.plex>

I look forward to receiving your revised manuscript.

Yours sincerely,

Zeljko Durdevic

We require:

- 1) A .docx formatted version of the manuscript text (including legends for main figures, EV figures and tables). Please make sure that the changes are highlighted to be clearly visible.
- 2) Individual production quality figure files as .eps, .tif, .jpg (one file per figure). For guidance, download the 'Figure Guide PDF': (<https://www.embopress.org/page/journal/17574684/authorguide#figureformat>).
- 3) A .docx formatted letter INCLUDING the reviewers' reports and your detailed point-by-point responses to their comments. As part of the EMBO Press transparent editorial process, the point-by-point response is part of the Review Process File (RPF), which will be published alongside your paper.
- 4) A complete author checklist, which you can download from our author guidelines (<https://www.embopress.org/page/journal/17574684/authorguide#submissionofrevisions>). Please insert information in the checklist that is also reflected in the manuscript. The completed author checklist will also be part of the RPF.
- 5) Please note that all corresponding authors are required to supply an ORCID ID for their name upon submission of a revised manuscript.
- 6) It is mandatory to include a 'Data Availability' section after the Materials and Methods. Before submitting your revision, primary datasets produced in this study need to be deposited in an appropriate public database, and the accession numbers and

database listed under 'Data Availability'. Please remember to provide a reviewer password if the datasets are not yet public (see <https://www.embopress.org/page/journal/17574684/authorguide#dataavailability>).

12) Author contributions: You will be asked to provide CRediT (Contributor Role Taxonomy) terms in the submission system. These replace a narrative author contribution section in the manuscript.

13) A Conflict of Interest statement should be provided in the main text.

14) Every published paper now includes a 'Synopsis' to further enhance discoverability. Synopses are displayed on the journal webpage and are freely accessible to all readers. They include a short stand first (maximum of 300 characters, including space) as well as 2-5 one-sentences bullet points that summarizes the paper. Please write the bullet points to summarize the key NEW findings. They should be designed to be complementary to the abstract - i.e. not repeat the same text. We encourage inclusion of key acronyms and quantitative information (maximum of 30 words / bullet point). Please use the passive voice. Please attach

these in a separate file or send them by email, we will incorporate them accordingly.

15) Include a Reagents and Tools Table as part of the Methods section, which can be downloaded from our author guidelines (<https://www.embopress.org/page/journal/17574684/authorguide#structuredmethods>)

***** Reviewer's comments *****

Referee #1 (Remarks for Author):

In this paper, the authors propose a novel in vitro model to study cerebral malaria pathogenesis and the role of pericytes. Ang1 and Tie2 are important molecules involved in endothelial cell - pericyte interaction and they have also assessed other markers such as N-cad, TIMP-1 to show impaired endothelial and pericyte connections, which is good. Conceptually, though, it should be noted that pericytes, in the absence of pre-existing (or malarial-induced) vascular injury, would never "see" the infected RBC. iRBC would need such an injury to "egress" microvessels. Critically, the iRBC egress products, the iRBC-egress media should be defined. This looks like uncharacterized supernatants. This should at least be addressed in the Discussion. The 3D model has limitations, such as being a model for larger vessels than capillaries, which they addressed in the Discussion. Results are interesting showing differences with Ang-1 treatment in vitro, but how would this be relevant in vivo? A mouse study to complement the results would have made the paper stronger. In the discussion, I feel like this could have been addressed better. Also, is recombinant ang-1 a realistic treatment? Can this be used directly as a treatment or would it require a carrier for delivery? This comment may be very clinical... they have suggested AKB-9778 could be the treatment used in the clinic, could they have used it in this model as proof of concept? Pericytes are recognised by more than 1 marker so it's great that the authors used 2-3 antibodies to detect pericyte coverage. However, the ratio of endothelial cells to pericytes in the brain is higher than in their model; in the brain its about 1-3 endothelial cells for 1 pericytes. I wonder if they tested this but found 5:1 to be better in their model... Realistically this model more closely replicates blood vessels in the kidney or lung where vessels are naturally more permeable. This limitation would need to be addressed in the Discussion. In addition to FITC dextran permeability, vascular integrity could be assessed by TEER. The authors only used one test for permeability and the scanning electron microscopy images don't show big changes in endothelial cell and pericyte connections. An endothelial cell control should have been included too.

Referee #2 (Remarks for Author):

Long et al. developed an advanced 3D microfluidics-based brain microvessel model that integrates human primary brain microvascular endothelial cells (HBMEC) and human brain vascular pericytes (HBVP). This model can successfully replicate a proper endothelial monolayer with approximately 30% pericyte coverage, which closely resembles the vascular structure of human cerebral arterioles. The authors use this system to study cerebral malaria (CM) pathophysiology with a focus to recapitulate the minimal functional cellular unit of the angiotensin-Tie signaling axis. Upon exposure to Plasmodium falciparum-infected red blood cell (iRBC)-egress products, the 3D microvessels exhibit increased endothelial permeability, significant reduction of pericyte branching, and decreased secretion of angiotensin-1 (Ang-1) into the culture medium, showing pericyte dysregulation as a key pathological feature. Additionally, while the authors found that Ang-2 secretion is reduced, the Ang-2:Ang-1 ratio remains significantly elevated. Pre-treatment with recombinant Ang-1 (18 hours) or the VE-PTP inhibitor AKB-9778 (1 hour) partially restores monolayer permeability. While the paracrine regulation of pericytes on endothelial cells via the angiotensin-Tie signaling axis is well-established, the authors should be commended for developing this 3D microvessel model, which uniquely enables the isolated study of direct endothelial-pericyte interactions. By faithfully replicating key cellular and structural features of the brain microvasculature, this model can certainly serve as a valuable platform for screening potential therapeutic interventions. The manuscript is well-written, and the experiments are clearly presented. However, a more in-depth analysis would further strengthen the study and enhance its impact.

Major points:

1. Cell type and 3D microvessel characterization: The integration of HBMEC into the 3D microvessel model is a key feature of this study. While Figure 1D demonstrates the presence of VE-cadherin-positive junctions, the characterization of the endothelial cells in Figure EV1 appears sub-optimally performed. Proper analysis of tight and adherens junction proteins relies on the formation of a continuous endothelial monolayer, raising concerns about the interpretation of Claudin-5 and VE-cadherin expression if junctions are not fully formed or intact. Additionally, the reported Claudin-5, VE-cadherin, and CD31 staining shows unexpected nuclear localization, which suggests potential non-specificity and requires further clarification. Brain endothelial barrier function is primarily regulated by Claudin-5, rather than VE-cadherin. This observation aligns with Figure 3D, where VE-cadherin expression and localization appear largely unaffected. Given the importance of Claudin-5 in brain EC tight junction integrity, the authors should perform additional staining and quantification of both junctional regulators to strengthen their conclusions. The observed increase in permeability to 70 kDa dextran further suggests that the endothelial barrier disruption is likely driven by Claudin-5 dysregulation. This aspect deserves greater attention, as BBB dysfunction in the human brain often involves subtle, progressive junctional changes, rather than an abrupt breakdown. Since a complete loss of BBB integrity would quickly result in severe pathology or fatal outcomes, a more thorough analysis of tight junction stability in this model would provide more physiological relevance and strengthen conclusions.

2. Serial block-face scanning electron microscopy: The images suggest that pericytes appear to be more confined to the corners of the microvessel structure, raising the question of whether this artificial localization could impact their functional interaction with endothelial cells. The authors should clarify whether this spatial arrangement affects pericyte coverage, signaling, or mechanical support of the endothelium. Additionally, to provide a more comprehensive structural assessment, the authors should include transverse sections of Figure 3E, which would help visualize the spatial organization of pericytes relative to the endothelial monolayer. This would further support the characterisation of cell-cell interactions within the 3D microvessel model.

3. Peg-and-socket structures: An important aspect of endothelial-pericyte communication involves peg-and-socket interactions, which facilitate direct contact and signaling between the two cell types. It would be highly informative to examine whether these structures are present in this model. If so, their distribution and frequency should be reported.

4. The angiopoietin-Tie axis: The authors report that pre-treatment with recombinant Ang-1 for 18 hours led to an 80% recovery in barrier function (line 351), while a 1-hour pre-treatment with AKB-9778 reduced microvessel leakage by 39% (lines 371-372) following exposure to iRBC-egress media. However, the images of 70 kDa FITC-dextran flux in Figure 5B+E appear to show less leakage in AKB-9778 pre-treated microvessels. The quantification data presentation is not entirely clear, and statistical details are missing.

Additionally, the observation that recombinant Ang-1 alone increases monolayer permeability is unexpected and requires further explanation. Could this be due to dose-dependent effects, receptor saturation, or altered pericyte-endothelial signaling? To gain deeper insights into the molecular pathways and potential transcriptional regulation involved in barrier restoration, the authors should analyze mRNA expression changes in both endothelial cells and pericytes following Ang-1 and AKB-9778 stimulation. If separation of these two cell populations cannot be guaranteed, scRNAseq could provide valuable info to distinguish cell type-specific responses.

5. Furthermore, in lines 427-428, the authors list several molecules present in the infected medium, but their specific impact on pericyte function remains unclear. A deeper understanding of these interactions may also help explain the observed reduction in Angiopoietin-2 levels.

Minor points:

1. What is known about Tie-1 expression in this co-culture model?

2. To clarify the source of Ang-2, the authors should repeat the experiment presented in Figure EV1D, this time analyzing Ang-2 expression. Can the authors comment on the seemingly uneven endothelial cell distribution across the microvessel in Figure 1B?

3. In Figure EV3 the authors explain how pericyte morphological features have been analysed. However, the method to count branches is not clearly explained in the methods section. Since the 3D segmented pericyte meshes show dynamic structures with very short branch outlines, please clarify how the branches were separated and counted.

Referee #3 (Comments on Novelty/Model System for Author):

Technically this kind of model has been used for vessel models and permeability and immunostaining are common methods to evaluate. Therefore, technical impact is not high. However, the vessel models are not widely used for infection models based on

my understanding. It might be a mere novelty of the manuscript.

Referee #3 (Remarks for Author):

The manuscript presents a well-structured and insightful study on the role of pericytes in CM pathogenesis. The work provides compelling evidence for the impact of *P. falciparum* egress products on the angiopoietin-Tie axis and vascular integrity using a microfluidic-based 3D brain microvascular model. The study is well-executed and contributes to the field of malaria research by uncovering a previously underappreciated role of pericytes, which also link to researches in on-chip vessel community. The manuscript can be publishable with minor revisions.

Major comments:

1. The study effectively demonstrates the impact of *P. falciparum* egress products on pericyte-mediated Ang-1 secretion and vascular integrity. However, the novelty of the work compared to prior studies using such on-chip vessel models for infection studies could be more explicitly discussed.
2. Although "3D brain microvessel model fabrication" was included in method section, details of the fabricated model are not graphically presented. Include some of top view, cross-sectional view, dimensions, and process flow.

Minor comments:

1. Fig. EV3 is not cited in the main text.
2. Limitations are discussed clearly. However, I recommend to discuss other types of on-chip vessels that enable monitoring of long-term vascular damage.

Rebuttal:

Referee #1 (Remarks for Author):

1. In this paper, the authors propose a novel in vitro model to study cerebral malaria pathogenesis and the role of pericytes. Ang1 and Tie2 are important molecules involved in endothelial cell - pericyte interaction and they have also assessed other markers such as N-cad, TIMP-1 to show impaired endothelial and pericyte connections, which is good.

We thank the reviewer for their positive feedback.

Conceptually, though, it should be noted that pericytes, in the absence of pre-existing (or malarial-induced) vascular injury, would never "see" the infected RBC. iRBC would need such an injury to "egress" microvessels.

Thank you for the suggestion. Clinical neuroimaging and postmortem data from cerebral malaria patients have extensively exemplified blood-brain barrier disruption through MRI measurements and histological evidence of leakage of large proteins into the brain parenchyma, such as fibrinogen (Seydel *et al*, 2015; Sahu *et al*, 2021; Dorovini-Zis *et al*, 2011). Following a suggestion from reviewer 2, our new results using 2D HBMEC monolayers suggest that *P. falciparum* iRBC-egress media causes a decrease in protein expression of a key regulator of the blood-brain barrier, claudin-5, shown now in Figure 3D. This result is in agreement with our previously described increase in microvessel permeability to 70 kDa-dextran in 3D brain microvessels upon iRBC-egress media incubation shown in Figure 3B. As 70 kDa is larger than the molecular weights of several potential disruptors of pericyte activity such as heme (outlined in the next comment below), as well as other parasite products, it is a possibility that iRBC egress products can pass through the described disrupted endothelial junctions and come into direct contact with pericytes.

Our group has also generated extensive transcriptomic data published on the effect of egress consistent with our current project products on endothelial cells that are consistent with our current project. These include a recent published manuscript on a human primary blood-brain barrier model (Piatti *et al*, 2025) and in a pre-print with iPSC differentiated endothelial cells (Korbmacher *et al*, 2025). The two transcriptomic analyses have further shown that egress products cause a robust downregulation in tight and adherens junction transcripts. Additionally, we identified more than 1000 differentially expressed genes in pericytes, including a minor

downregulation of Ang-1 in response to iRBC egress products (Piatti *et al*, 2025). Together, the transcriptomic studies and the current study propose a mechanism by which *P. falciparum*-egress products cause endothelial breakdown and leak into the perivascular space affecting pericyte secretion of Ang-1, which in turn further exacerbates endothelial barrier breakdown. However, an unresolved question remains whether the dysregulation seen in pericytes are a direct effect of egress products following initial endothelial barrier disruption or/and as a result of dysregulated cross-talk with endothelial cells. Although determination of the exact mechanism behind pericyte dysregulation is outside of the scope of this project, directed analysis of the transcriptomic data we previously generated (Piatti *et al*, 2025) has highlighted that BTB domain and CNC homolog 1 (Bach-1) is significantly upregulated in pericytes following incubation with iRBC-egress media. Notably, Bach-1 is a transcriptional repressor of Ang-1 previously shown to be activated by heme (Slater *et al*, 2018). While this finding highlights pericyte interaction with released heme as a potential inducer of Ang-1 dysregulation, future studies will be required to better explore this hypothesis.

To address the reviewer's comment, we have included the following section in the discussion.

Line 520: *"To our knowledge, our study is the first one to report that P. falciparum products released upon iRBC egress halt Ang-1 protein secretion by pericytes. This is in line with a recent single cell RNA sequencing analysis from our group showing decreased transcriptional expression of Ang-1 in pericytes and Tie-2 (TEK) in endothelial cells following iRBC-egress media incubation (Piatti et al., Biorxiv). Nevertheless, an unresolved question remains whether pericyte dysregulation is directly mediated by parasite egress products crossing the vascular barrier, or through disrupted endothelial-pericyte crosstalk."*

Critically, the iRBC egress products, the iRBC-egress media should be defined. This looks like uncharacterized supernatants.

This should at least be addressed in the Discussion.

We agree with the reviewer that it would be interesting to characterize the active components of the iRBC-egress media. However, it is quite likely that multiple previously characterized disruptive components of the iRBC-egress media could act simultaneously to promote the junctional and pericyte dysregulation characterized in this study, and therefore these components require isolation and addition following a systematic approach to examine their individual impacts within the model. As mentioned in the manuscript, these include parasite proteins (histones, histidine-rich proteins (HRP)), human proteins (heme, histones), glycolipids (GPI-anchors), and a complex mixture of organic-inorganic compounds (hemozoin). Because this project would require a deep molecular characterization of multiple parasite components in a low throughput system, we consider it is outside of the scope of the current project. Nevertheless, as suggested by the reviewer, we have addressed this topic in the discussion, highlighting evidence of potential interactions between iRBC-egress products and pericytes responsible for their dysregulation.

To address all these relevant points, we have added to multiple sections of the discussion:

Line 487: *“P. falciparum products released during blood stage egress have been shown to possess barrier breakdown capabilities on 2D endothelial monolayers grown on plastic. Proposed barrier disruptive P. falciparum-iRBC egress products include merozoite proteins, histones, heme, hemozoin and P. falciparum histidine rich protein 2 (Gomes et al, 2023; Gillrie et al, 2007; Gallego-Delgado et al, 2016; Pal et al, 2016; Storm et al, 2019; Moxon et al, 2020; Zuniga et al, 2022). Our studies in 2D HBMEC monolayers showed, for the first time, that exposure to iRBC-egress products induces barrier breakdown as a consequence of a decrease in protein expression of the tight junctional marker Claudin-5”*

Line 527: *“Furthermore, future studies are necessary to isolate and identify which of the multiple iRBC egress components are responsible for altered signaling pathways in the microvasculature. Although most of them have shown strong disruptive effects when added separately to 2D endothelial monolayers, it remains unclear whether they will exert the same degree of damage once isolated as observed following iRBC-egress media incubation in this 3D model with pericyte co-culture and enhanced barrier properties.*

2. The 3D model has limitations, such as being a model for larger vessels than capillaries, which they addressed in the Discussion.

Results are interesting showing differences with Ang-1 treatment in vitro, but how would this be relevant in vivo? A mouse study to complement the results would have made the paper stronger. In the discussion, I feel like this could have been addressed better.

We agree with the reviewer that the functional and therapeutic role of the angiotensin-Tie axis in cerebral malaria has been extensively explored through use of mouse models (Higgins *et al*, 2016). Importantly to this study, altered Ang-1 and Ang-2 serum levels found in cerebral malaria patients have been successfully recapitulated in mouse models (Higgins *et al*, 2016). Therefore, future therapeutic strategies targeted to pericyte function and the angiotensin-Tie axis identified in this study, such as AKB-9778, could be explored and validated in such animal models. Hence, we have added these references to the discussion to better highlight the previous findings in mouse model studies. However, translation of findings in mouse experimental malaria models to human studies have so far been challenging. For example, inhalation of nitric oxide in a cerebral malaria experimental mouse model showed improved vascular integrity, including angiotensin-Tie parameters, and 100% survival (Serghides *et al*, 2011), yet no

improvements on clinical outcome or angiopoietin levels were observed in a randomized clinical trial (Hawkes *et al*, 2015). The main difference between *P. falciparum* human infection and experimental cerebral malaria mouse models is that rodent malaria parasites present limited accumulation in the microvasculature. Our results show that the addition of egress products, as well as, perfusion and sequestration of *P. falciparum*-iRBC in our microvascular model had similar effects on the angiopoietin-Tie pathway as observed in experimental cerebral malaria mouse models. Furthermore, our bioengineered approach provides additional mechanistic insights into disease pathogenesis. For example, we have first identified that 1) Ang-1 depletion can be caused by pericyte alteration, and 2) that parasite products are directly responsible for ang-1 depletion, two findings revealed through the simplified bioengineered 3D microvascular model that could have been hard to discern in the complexity of an animal model. Therefore, our *in vitro* studies have allowed for a deeper understanding of angiopoietin-Tie axis disruption and suggested pericytes as a novel therapeutic target for CM treatment. Although no one model is perfect, we agree with the reviewer that complementary experimental approaches in animal models and new bioengineered approaches would be beneficial in the long-term for cerebral malaria research.

We have now added to the discussion:

Line 554: *“Previously, administration of a recombinant Ang-1 tetramer protein with improved stability, BowAng1, has shown improved outcomes in a rodent experimental CM model even after onset of disease, preserving blood-brain barrier integrity, and leading to a significantly increased survival rate (Higgins et al, 2016). However, translation of therapeutics identified in rodent experimental CM models to the clinic have proven challenging. For example, inhalation of nitric oxide, an activator of the angiopoietin-Tie axis (Zacharek et al, 2006), was shown to be successful in rodent experimental CM models (Serghides et al, 2011) but showed no improvement in a human clinical trial (Hawkes et al, 2015). Therefore, we believe that future combination of experimental approaches using 3D bioengineered microvessels with mouse P. berghei models would enhance translation and development of adjunctive therapies, improving care of CM patients.”*

3. Also, is recombinant ang-1 a realistic treatment? Can this be used directly as a treatment or would it require a carrier for delivery? This comment may be very clinical... they have suggested AKB-9778 could be the treatment used in the clinic, could they have used it in this model as proof of concept?

First of all, we would like to indicate that characterization of the therapeutic ability of AKB-9778 within the model was already included in the previous version of the manuscript. We apologized if the wording was not clear, it can be now found in current Figure 6F. The use of recombinant angiopoietin-1 in this study was to highlight the importance that decreased angiopoietin-1 secretion could have on malaria-induced barrier breakdown, representing a proof of concept that this pathway can be a potential therapeutic treatment of disease. We have further clarified this point in the results section:

Line 398: *“Taken together, pre-treatment with rAng-1 partially protects against the microvascular permeability caused by iRBC egress products, highlighting the importance that a decrease in Ang-1 secretion could have in exacerbating CM-induced vascular disruption, as well as,*

showing a potential for vascular restorative pathways that target the angiotensin-Tie2 axis in CM treatment.”

We agree with the reviewer that recombinant Ang-1 itself is likely not a realistic treatment as the protein possesses a complex multi-domain structure that promotes aggregation and insolubility, hindering large-scale recombinant production (Wallace *et al*, 2021). To address this problem, several modified recombinant angiotensin-1 variants have been generated, providing protein stability while maintaining potent Tie-2 activation. These include COMP-Ang1 (Cho *et al*, 2004) and BowAng1 (Davis *et al*, 2003), with the latter being successfully tested in an experimental cerebral malaria model (Higgins *et al*, 2016). As far as we know, these recombinant proteins have not been tested in clinical trials and still require further research into potential adverse effects during patient administration (Wallace *et al*, 2021). We instead chose to target Tie-2 activation by AKB-9778 as a therapeutic treatment in our model (Figure 6F), since this compound has been well tolerated in patients and is currently in phase II clinical trial for other diseases associated with angiotensin-Tie axis and pericyte dysregulation, such as diabetic macular edema (Campchiaro *et al.*, 2016) and diabetic hypertension (Siragusa *et al.*, 2021). Therefore, AKB-9778 better represents a potential option for clinical cerebral malaria treatment than recombinant angiotensin-1 variants. Indeed, AKB-9778's ability to rapidly act (requiring only a 1h pre-incubation) (EV6) compared to recombinant Ang-1 (18h pre-incubation) (EV5), increases its potential as a rapid-acting therapeutic for severe patients already displaying symptoms of CM, further underlining our model as a valuable platform to identify potential therapeutics to counteract malaria-induced brain vasculature breakdown.

4. Pericytes are recognised by more than 1 marker so it's great that the authors used 2-3 antibodies to detect pericyte coverage. However, the ratio of endothelial cells to pericytes in the brain is higher than in their model; in the brain its about 1-3 endothelial cells for 1 pericytes. I wonder if they tested this but found 5:1 to be better in their model... Realistically this model more closely replicates blood vessels in the kidney or lung where vessels are naturally more permeable. This limitation would need to be addressed in the Discussion.

Upon development of the brain microvasculature model, we tested several different ratios of endothelial cells to pericytes to determine the optimal conditions that best replicate the mentioned 1-3 endothelial cell to 1 pericyte ratio observed in the vasculature of the brain (Sims, 1984; Mathiisen *et al.*, 2010). We have now included this initial experiment in Figure EV2D. This includes microvessel seeding of both the 3:1 endothelial cell to pericyte ratio present in the brain microvasculature and the 5:1 ratio used in this study. While the 3:1 endothelial cell to pericyte ratio resulted in the presence of an extensive pericyte sheath around the endothelial microvessels, these microvessels displayed significant contraction after 3 days in culture leading to detachment from the collagen hydrogel and microvessel lumen collapse. Alternatively, use of a 5:1 endothelial cell to pericyte ratio resulted in the formation of microvessels that maintained a perfusable lumen similar to that of endothelial-only microvessels while possessing the previously reported *in vivo* 30% endothelial surface coverage by pericytes (Fig. 1B,C). Therefore, we decided to perform all of our experiments utilizing the 5:1 endothelial cell to pericyte ratio.

Figure EV2D:

We therefore added these results to better describe the model development, as well as, the following sentences in the results and the discussion.

Results:

Line 168: “After 3 days in culture, the two cell types reorganized into two different layers with HBMEC forming a network of 100- μ m diameter microvessels with HBVP covering the microvessel surface (Figs. 1B and EV2B). To optimize the ratio of endothelial cells to pericytes required to produce perfusable microvessels with high pericyte coverage, we characterized the use of both a 3:1 and 5:1 endothelial cell to pericyte seeding ratio. The 3:1 HBMEC:HBVP ratio resulted in microvessels which displayed significant contraction, resulting in microvessel lumen collapse. In contrast, the 5:1 HBMEC:HBVP ratio formed viable and lumenized microvessels, and therefore, was chosen for subsequent studies (Fig. EV2D).”

Discussion:

Line 471: Here, we fabricated an *in vitro* model of the human brain microvasculature by co-seeding primary brain endothelial cells and pericytes through the microfluidic network. Although vessels simultaneously seeded with a 3:1 endothelial cell:pericyte ratio were not viable, a 5:1 ratio reproduced the high pericyte coverage found in the microvasculature of the human brain.

Line 479: *Despite these advancements, the ratio of endothelial cells to pericytes within the model was still higher than the 3:1 to 1:1 endothelial: pericyte ratio reported in vivo (Sims, 1984; Mathiisen et al., 2010). Therefore, future improvements to the model could include concurrent addition of pericytes to the collagen bulk to increase the number of pericytes within the model without inducing extensive microvessel contraction.*

5. In addition to FITC dextran permeability, vascular integrity could be assessed by TEER. The authors only used one test for permeability and the scanning electron microscopy images don't show big changes in endothelial cell and pericyte connections

While TEER assessment represents a non-invasive method to measure vascular integrity in real-time, conducting TEER measurements in the brain microvasculature model remain difficult due to the impossibility of placing one electrode inside of a 100-200 μm diameter microvessel and a second one in the hydrogel bulk. Furthermore, TEER measurement requires the presence of an even electrical field, which is difficult to control within variable hydrogels such as collagen (Herland *et al*, 2016). Therefore, we utilized FITC dextran permeability in this study, as this allows comparison of our model with previous brain microvasculature models, as well as, *in vivo* benchmarks. Additionally, each of our FITC dextran permeability studies were complemented with xCELLigence studies on HBMEC monolayers, which while simpler in biophysical properties and lacking pericytes, provided us with real-time barrier measurements similar to those acquired in TEER studies.

An endothelial cell control should have been included too.

As suggested by the reviewer, we also included a comparison of 70 kDa FITC dextran permeability values between the brain microvasculature model and brain endothelial-only devices (Fig. 1F). As observed in previous studies, the presence of brain pericytes led to a non-significant decrease in the permeability of the microvessels (Herland *et al*, 2016).

Line 182: *“Consistent with the presence of continuous endothelial junctions, the permeability of 70 kDa FITC dextran was decreased in microvessels containing pericytes (median= 1.23x10⁻⁶*

cm/s) compared to HBMEC only microvessels (median= 1.83×10^{-6} cm/s), albeit not significantly (Fig. 1F)."

This could be due to the absence of astrocytes in our model which has been shown to decrease permeability in other 3D vascular models (Herland *et al*, 2016; Hajal *et al*, 2022; DeStefano *et al*, 2018). Although our group has generated devices with astrocytes, in this study we deliberately only included the main cellular components required to replicate the angiotensin-Tie axis to examine the specific role of brain pericytes during CM pathogenesis.

Discussion:

Line 458: *"Here, we developed a 3D microfluidic brain microvessel model that recapitulates the minimal functional brain microvascular unit to recreate the angiotensin-Tie axis, including both primary brain microvascular endothelial cells and pericytes."*

Line 604: *"Additionally, cell types such as astrocytes have been shown to further aid in strengthening the barrier properties of in vitro models, as the permeability displayed here remained supraphysiological to values observed in vivo (DeStefano *et al*, 2018). Yet, one of the major advantages of in vitro bioengineered models is the opportunity to sequentially introduce different components that, independently or collectively, could play a role in a complex and multifaceted disease such as CM. Subsequent iterations of the model presented here could introduce other cell types that produce Ang-1 or incorporate pro-inflammatory cytokines or thrombin, to concurrently model functional consequences of increased secretion of Ang-2 by endothelial cells."*

Referee #2 (Remarks for Author):

Long *et al.* developed an advanced 3D microfluidics-based brain microvessel model that integrates human primary brain microvascular endothelial cells (HBMEC) and human brain vascular pericytes (HBVP). This model can successfully replicate a proper endothelial monolayer with approximately 30% pericyte coverage, which closely resembles the vascular structure of human cerebral arterioles. The authors use this system to study cerebral malaria (CM) pathophysiology with a focus to recapitulate the minimal functional cellular unit of the angiotensin-Tie signaling axis. Upon exposure to *Plasmodium falciparum*-infected red blood cell (iRBC)-egress products, the 3D microvessels exhibit increased endothelial permeability, significant reduction of pericyte branching, and decreased secretion of angiotensin-1 (Ang-1) into the culture medium, showing pericyte dysregulation as a key pathological feature.

Additionally, while the authors found that Ang-2 secretion is reduced, the Ang-2:Ang-1 ratio remains significantly elevated. Pre-treatment with recombinant Ang-1 (18 hours) or the VE-PTP inhibitor AKB-9778 (1 hour) partially restores monolayer permeability.

While the paracrine regulation of pericytes on endothelial cells via the angiotensin-Tie signaling axis is well-established, the authors should be commended for developing this 3D microvessel model, which uniquely enables the isolated study of direct endothelial-pericyte interactions. By faithfully replicating key cellular and structural features of the brain microvasculature, this model can certainly serve as a valuable platform for screening potential therapeutic interventions. The manuscript is well-written, and the experiments are clearly presented. However, a more in-depth analysis would further strengthen the study and enhance its impact.

We would like to thank the reviewer for the positive feedback. We have addressed the main major and minor comments below.

Major points:

1. Cell type and 3D microvessel characterization: The integration of HBMEC into the 3D microvessel model is a key feature of this study. While Figure 1D demonstrates the presence of VE-cadherin-positive junctions, the characterization of the endothelial cells in Figure EV1 appears sub-optimally performed. Proper analysis of tight and adherens junction proteins relies on the formation of a continuous endothelial monolayer, raising concerns about the interpretation of Claudin-5 and VE-cadherin expression if junctions are not fully formed or intact. Additionally, the reported Claudin-5, VE-cadherin, and CD31 staining shows unexpected nuclear localization, which suggests potential non-specificity and requires further clarification.

Thank you for correctly raising this concern regarding sub-optimal formation of endothelial junctions within our 2D HBMEC monolayers. To perform high-resolution imaging to sufficiently examine the localization of proteins within the cells in 2D, we grow HBMECs on glass 8-well IBIDI chambers which, despite coating, can result in incomplete monolayer formation in our hands. Therefore, we repeated the 2D HBMEC characterization experiments only using wells where complete monolayer formation occurred (Figure EV1A,B). Upon correct monolayer formation, where no gaps could be seen, Claudin-5, VE-cadherin and CD31 displayed clear junctional labeling with minimal nuclear localization (Figure EV1A,B).

Brain endothelial barrier function is primarily regulated by Claudin-5, rather than VE-cadherin. This observation aligns with Figure 3D, where VE-cadherin expression and localization appear largely unaffected. Given the importance of Claudin-5 in brain EC tight junction integrity, the authors should perform additional staining and quantification of both junctional regulators to strengthen their conclusions. The observed increase in permeability to 70 kDa dextran further suggests that the endothelial barrier disruption is likely driven by Claudin-5 dysregulation. This aspect deserves greater attention, as BBB dysfunction in the human brain often involves subtle, progressive junctional changes, rather than an abrupt breakdown. Since a complete loss of BBB integrity would quickly result in severe pathology or fatal outcomes, a more thorough analysis of tight junction stability in this model would provide more physiological relevance and strengthen conclusions.

Thank you for the suggestion. To provide a more thorough understanding of how Claudin-5 and VE-cadherin are altered following egress media incubation we performed a western blot of cell lysates from 2D HBMECs monolayers incubated 18 hours with 5×10^7 ruptured iRBC/mL (iRBC-egress media) or a media-only control as done with the microvessel model. We choose to

conduct a western blot analysis on HBMEC monolayers because 1) Claudin-5 labeling has provided only low-quality staining in our 3D model with commercially available antibodies due to the challenges of performing an immunofluorescence under microfluidic conditions, 2) methanol fixation, which offers better preservation of claudin-5 epitopes for antibody labeling, is not compatible with our 3D microvessel model, and 3) only a relatively small number of cells can be extracted from each microvessel device and therefore it would require an unfeasible number of devices to extract enough cells to perform western blot analysis on microvessels. In 2D monolayers, Claudin-5 showed a significant decrease in total protein levels upon incubation with iRBC-egress media, suggesting a role of Claudin-5 dysregulation in increased microvessel permeability (Fig. 3D). On the contrary, VE-cadherin total protein levels did not change after iRBC-egress media incubation, in line with the suggestion by the reviewer and the minimal changes observed through fluorescence imaging (Fig. 3E). Overall, these results tie together our findings on iRBC-egress media-mediated barrier breakdown with disruption of important tight junctional proteins involved in proper BBB function.

We have modified the results section as follows:

Line 274: *“To better quantify changes in junction proteins upon barrier breakdown, we treated HBMEC monolayers with iRBC-egress media for 18 hours and examined total protein levels of claudin-5 and VE-cadherin by Western blotting. Upon iRBC-egress media incubation, a significant decrease in total claudin-5 protein levels was observed (Fig. 3D) while those of VE-cadherin remained unaltered (Fig. 3E), suggesting a role of tight junctional disruption in barrier breakdown by iRBC-egress media.”*

New Figure 3 subpanels:

We also have added to the discussion:

Line 492: *“Our studies in 2D revealed, for the first time in HBMEC, that exposure to iRBC-egress products induces barrier breakdown as a consequence of decreased protein expression of the tight junctional marker claudin-5, in agreement with the well-documented role of claudin-5 as a key regulator of blood-brain barrier breakdown in in vivo models of multiple sclerosis (Argaw et al, 2009; Paul et al, 2013), traumatic brain injury (Campbell et al, 2012) and depression (Sun et al, 2024).”*

2. Serial block-face scanning electron microscopy: The images suggest that pericytes appear to be more confined to the corners of the microvessel structure, raising the question of whether this artificial localization could impact their functional interaction with endothelial cells. The authors should clarify whether this spatial arrangement affects pericyte coverage, signaling, or mechanical support of the endothelium. Additionally, to provide a more comprehensive structural assessment, the authors should include transverse sections of Figure 3E, which would help visualize the spatial organization of pericytes relative to the endothelial monolayer. This would further support the characterisation of cell-cell interactions within the 3D microvessel model.

To clarify the distribution of pericytes around the endothelial microvessel, we have now included both a transversal cross-section image from the serial block-face scanning electron microscopy dataset, showing the segmentation of the endothelium (white) and the surrounding pericytes (coloured) (Figure 4C), and transversal cross-sectional views of the 3D-rendered pericytes from microvessels incubated with either media-only or iRBC-egress media (Figure 4D). Inclusion of these additional images should provide further clarity of pericyte distribution along the entire endothelial surface, not just restriction to the corners of the microvessel.

This distribution is in line with fluorescence analysis in Figure 1B,D and 1E where pericytes are shown to cover the ~30% of the top and bottom surfaces of the microvessels. The new transversal EM views seem to point to an increase in pericyte coverage in the iRBC-egress media condition. Nevertheless, SBF-SEM is a time consuming and expensive technique limited to a very small region of interest within the microvessel network of a single device, which is not ideal to assess total vessel coverage. Therefore, to further analyze potential changes in pericyte coverage, we have performed a new and deeper analysis on pericyte coverage in figure 4A with and without iRBC-egress media incubation. When we quantified pericyte coverage in a higher

number of microvessel branches imaged with confocal microscopy, we did not observe any difference between the control and the iRBC-egress media treated conditions. This result further highlights an absence of pericyte detachment or whole cell morphological changes in the presence of iRBC-egress media.

The new result section includes:

Line 305: *“To determine whether increased permeability was associated with physical changes in pericytes, we first assessed pericyte coverage of the microvessel surface. In the presence of iRBC-egress media, pericytes maintained coverage of the endothelium to a similar degree of that observed in media only-treated microvessels (Fig. 4A,B). Next, pericyte cellular morphology was examined at the ultrastructural level by SBF-SEM. In line with pericyte coverage analysis, HBVP were located around the entire microvessel perimeter (Fig. 4C,D) and maintained close spatial proximity to the endothelium in the presence of P. falciparum-iRBC ghosts, similar to control devices exposed to vascular growth media (Fig. 4D). To investigate potential changes in HBVP shape, we isolated and analyzed each 3D rendered HBVP (Figs. 4D,E and EV3D,E).”*

3. Peg-and-socket structures: An important aspect of endothelial-pericyte communication involves peg-and-socket interactions, which facilitate direct contact and signaling between the two cell types. It would be highly informative to examine whether these structures are present in this model. If so, their distribution and frequency should be reported.

Thank you for inquiring about peg-and-socket junctions within our model, as we too believe that they are an important aspect of endothelial-pericyte communication that would benefit from an *in vitro* model to better tease apart the molecular components that govern them during homeostatic and disease states. We initially examined the brain microvasculature model for the presence of peg-and-socket junctions by transmission electron microscopy and confirmed their presence (Panel A in images below). However, this batch of primary HBVP caused severe microvessel constriction and vessel regression, and therefore, we decided to perform all experiments of the current manuscript with another primary HBVP batch. However, no peg-and-socket junctions were found in the current batch of pericytes by SBF-SEM. This could be due to the lack of such structures in the region of interest imaged or an absence of peg-and-socket junction formation in this second batch. Unfortunately, as volume electron microscopy studies are extremely low throughput and time-consuming, we could not follow up on the discrepancies between the two HBVP batches and we prefer not to include the TEM images in our analysis.

However, we agree with the reviewer that modelling peg-and-socket junctions *in vitro* would be of interest in future studies.

A

Cross-section: HBVP batch #2

Pericyte batch dependent peg-and-socket junction formation. (A) The presence of peg-and-socket junctions in pericytes (denoted by yellow asterisks) and endothelial cells (denoted by blue asterisks) on 3D microvessels generated with a distinct HBVP batch. L= lumen, C= collagen, P= pericyte and EC= endothelial cell. Scale bars represent 500 nm.

4. The angiopoietin-Tie axis: The authors report that pre-treatment with recombinant Ang-1 for 18 hours led to an 80% recovery in barrier function (line 351), while a 1-hour pre-treatment with AKB-9778 reduced microvessel leakage by 39% (lines 371-372) following exposure to iRBC-egress media. However, the images of 70 kDa FITC-dextran flux in Figure 5B+E appear to show less leakage in AKB-9778 pre-treated microvessels. The quantification data presentation is not entirely clear, and statistical details are missing.

Thank you for asking for clarification on the quantification of the permeability data presented in the paper. We acknowledge that this experiment is quite complex, as pre-treatment with recombinant Ang-1 was done in serum-free media to exclude addition of external Ang-1 from fetal bovine serum to the microvessels. The presence of Ang-1 in serum was confirmed by comparing freshly made media with or without serum by Luminex measurement (EV4C, below).

EGM-2MV + growth factors

Figure EV4C

Line 385: “*rAng-1* pre-treatment experiments were done in the absence of serum, an external source of angiopoietin-1 (EV4C).”

Next, we have now included a graph to highlight how pre-treatment of the microvessels with serum-free media results in an increase in baseline microvessel permeability compared to those given serum-containing media (EV5G, below), a finding that we did not consider in the previous submission of this manuscript. Therefore, this serum-free media only condition was missing from the original manuscript, and we believe that its addition provides more clarity to the minor non-significant increase in permeability observed with the addition of *rAng-1*, evident in the previous version of this manuscript when compared to the permeability of microvessels pre-treated with serum-containing media (below). In this new graph, we show that this increase in permeability in microvessels treated with *rAng-1* likely is due to the use of serum-free media when performing *rAng-1* pre-treatment rather than to *rAng-1* itself, as *rAng-1* pre-treated microvessels do not differ significantly in permeability to those pre-treated with serum-free media only.

Figure EV5G

Line 390: “Next, we tested the protective capacity of *rAng-1* against *iRBC*-egress products in the 3D microvessel model (Fig. 6A) also in the absence of serum, which non-significantly increased the baseline permeability of resting microvessels.”

Furthermore, to fully address the reviewer’s comment, we have now corrected Figure 6C (below) by replacing the serum-containing media only control condition that was previously in the graph with the more appropriate serum-free media only control condition, as well as, including more replicates of *iRBC*-egress media following pre-treatment with either *rAng-1* or serum-free media only to better clarify the protective nature of *rAng-1*. Therefore, Figure 6C now accurately reflects an increase in baseline permeability and hence the observation of higher levels of 70 kDa FITC-dextran leakage in the *rAng-1* representative images compared to those of the AKB-9778 experiments, which were completely performed in serum-containing media as AKB-9778 acts downstream of Ang-1. Additionally, all corresponding quantification of permeability fold change and partial protection following *rAng-1* pre-treatment was re-done using this new serum-free media only condition as the baseline permeability value of untreated

devices (see section below on the calculation used). Our current results now give rAng-1 recovery values (36% protection rather than the previously stated 80%) that are more in agreement with the recovery values observed with AKB-9778 (35%) and our 2D HBMEC monolayer studies (40% partial protection for rAng-1 (EV5)).

Figure 6C

Lastly, in the current version we have clarified our quantification and statistical details in the text, as well as, included the median values of each condition within the results section to allow the reader to better visualize the trends.

Line 393: “Microvessels exposed to iRBC-egress media (median= 9.90×10^{-6} cm/s) presented a normalized 2.3 increase in permeability compared to a media only control (median= 4.39×10^{-6} cm/s), while microvessels exposed to iRBC-egress media and pre-treated with Ang-1 (median= 9.13×10^{-6} cm/s;) presented a normalized 1.5-fold increase in permeability compared to microvessels pre-treated with Ang-1 (median= 6.17×10^{-6} cm/s), representing a 36% protection against microvessel leakage after exposure to iRBC-egress media (Fig. 6B,C).”

Line 417: “Microvessels pre-treated for 1h with AKB-9778 and exposed to iRBC-egress media (median= 3.15×10^{-6} cm/s) presented a non-significant 1.3-fold increase in permeability compared to AKB-9778 pre-treated microvessels (median= 2.35×10^{-6} cm/s) while untreated microvessels exposed to iRBC-egress media (median= 5.90×10^{-6} cm/s) presented a significant 2-fold increase in permeability compared to a media only control (median= 2.97×10^{-6} cm/s) representing a 35% protection against microvessel leakage after exposure to iRBC-egress media (Fig. 6E,F). Furthermore, a significant decrease in permeability was observed in iRBC-egress media-exposed devices pre-treated with AKB-9778.”

Line 792: “To calculate the percentage of recovery with either rAng-1 or AKB-9778 we divided the median permeability values of rAng-1/AKB-9778 pre-treated microvessels in the presence of iRBC-egress media by the permeability values of control rAng-1/AKB-9778 pre-treated microvessels in the absence of iRBC products. These values were compared with the same normalization done in untreated devices with and without iRBC-egress media.”

Additionally, the observation that recombinant Ang-1 alone increases monolayer permeability is

unexpected and requires further explanation. Could this be due to dose-dependent effects, receptor saturation, or altered pericyte-endothelial signaling? To gain deeper insights into the molecular pathways and potential transcriptional regulation involved in barrier restoration, the authors should analyze mRNA expression changes in both endothelial cells and pericytes following Ang-1 and AKB-9778 stimulation. If separation of these two cell populations cannot be guaranteed, scRNAseq could provide valuable info to distinguish cell type-specific responses.

We agreed with the reviewer regarding the interest on the Ang-1 increased monolayer permeability in the 3D model. Nevertheless, we consider that caution should be taken on the overinterpretation of these results, given that the rAng-1 only condition showed no significant difference when compared to the newly included serum-free media only condition (EV5G). Furthermore, we believe that this experiment falls out of scope of the malaria focus of this work. For these two reasons, we preferred not to conduct the transcriptomic experiment proposed by the reviewer.

5. Furthermore, in lines 427-428, the authors list several molecules present in the infected medium, but their specific impact on pericyte function remains unclear. A deeper understanding of these interactions may also help explain the observed reduction in Angiopoietin-2 levels.

Rebuttal:

Thank you for the suggestion, as also suggested in the first comment from reviewer #1, we have expanded on potential interactions between iRBC-egress products and pericytes that could regulate angiopoietin-1 secretion in the discussion. In terms of the observed reduction in Angiopoietin-2 levels, we have chosen to leave discussion on this topic out of the text as previous studies on known components of iRBC-egress media and angiopoietin-2 expression have been conflicting. For example, histones and HRP2 have been demonstrated to increase angiopoietin-2 expression (Yin *et al*, 2025; Harbuzariu *et al*, 2022), while mediators of heme-induced ferroptosis have been shown to decrease angiopoietin-2 by activating transcription factor, E2F1 (Warnatz *et al*, 2011). Therefore, to avoid inclusion of uncertain hypotheses on the reduction of angiopoietin-2 observed in our studies we have simply compared our findings to other studies (Gomes *et al*, 2023).

Minor points:

1. What is known about Tie-1 expression in this co-culture model?

Thank you for the suggestion. We have tried to stain HBMECs with an anti-Tie-1 antibody, however, the staining appeared to be non-specific in our hands. Therefore, we prefer not to focus on Tie-1 in this paper and will instead further characterize it in future studies.

2. To clarify the source of Ang-2, the authors should repeat the experiment presented in Figure EV1D, this time analyzing Ang-2 expression.

Rebuttal:

Thank you for the suggestion. We have now included analysis of Ang-2 secretion (Figure EV1E) similar to that of Ang-1, suggesting that endothelial cells secrete Ang-2 with increased secretion in the presence of pericytes.

The result section was modified as follows:

Line 156: *“Furthermore, we confirmed exclusive secretion of Ang-1 by HBVP and HBMEC secretion of Ang-2 in a transwell model (Fig. EV1D).”*

Can the authors comment on the seemingly uneven endothelial cell distribution across the microvessel in Figure 1B?

The uneven endothelial cell distribution in Figure 1B is an imaging artifact that occurs as our devices often present a slight tilt when being imaged. Noteworthy, this is quite a large tilescan, being 4x4mm in dimension. Therefore, tilescans of the microvessel network often appear with higher immunofluorescence intensity on one side of the device versus the other due to increased imaging depth being necessary for the side tilted farthest from the objective. To exemplify the homogenous endothelial coverage across the entire device, we have included below the corresponding Z-projection binary mask of vWF-labeled HBMECs showing equal cell distribution across the microvessel network.

3. In Figure EV3 the authors explain how pericyte morphological features have been analysed. However, the method to count branches is not clearly explained in the methods section. Since the 3D segmented pericyte meshes show dynamic structures with very short branch outlines, please clarify how the branches were separated and counted.

Thank you for the suggestion. Pericyte branches were identified as cellular protrusions branching from the main cell body that were greater than 10 μm in length. Choice of a 10 μm threshold allowed for stringent analysis of pericyte morphology in accordance to previous studies characterizing *in vivo* pericyte morphology, where pericyte branches were determined to be greater than 10 μm in length (Hartmann et al., 2015).

Line 763: *“Pericytic branches were defined as cellular protrusions greater than 10 μm in length that bifurcate from the main cell body, as described previously (Hartmann et al., 2015).”*

Referee #3 (Comments on Novelty/Model System for Author):

Technically this kind of model has been used for vessel models and permeability and immunostaining are common methods to evaluate. Therefore, technical impact is not high. However, the vessel models are not widely used for infection models based on my understanding. It might be a mere novelty of the manuscript.

Referee #3 (Remarks for Author):

The manuscript presents a well-structured and insightful study on the role of pericytes in CM pathogenesis. The work provides compelling evidence for the impact of *P. falciparum* egress products on the angiopoietin-Tie axis and vascular integrity using a microfluidic-based 3D brain microvascular model. The study is well-executed and contributes to the field of malaria research by uncovering a previously underappreciated role of pericytes, which also link to researches in on-chip vessel community. The manuscript can be publishable with minor revisions.

We thank the reviewer for the support on the manuscript.

Major comments:

1. The study effectively demonstrates the impact of *P. falciparum* egress products on pericyte-mediated Ang-1 secretion and vascular integrity. However, the novelty of the work compared to prior studies using such on-chip vessel models for infection studies could be more explicitly discussed.

Thank you for the suggestion. To better highlight the novelty of our model to examine the interaction of endothelial cells and pericytes in the context of infectious disease we have added the following sentence to the discussion:

Line 467: *“In vitro microfluidics-based brain vascular models have been recently utilized in the study of several infectious diseases, including those caused by malaria (Howard et al, 2023; Bernabeu et al, 2019), neurotropic viruses (Zhang et al, 2025; Wang et al, 2024) and fungi (Kim et al, 2021). However, most of these models do not include pericytes or lack detailed characterization of brain endothelial-pericyte interactions.”*

2. Although "3D brain microvessel model fabrication" was included in method section, details of the fabricated model are not graphically presented. Include some of top view, cross-sectional view, dimensions, and process flow.

Thank you for this suggestion. We have included a graphical figure of model fabrication in Figure EV2A,B and C. This figure includes detailed dimensional information, a schematic of top

and cross-sectional view, as well as, representative images of the two microfluidic networks used in this study.

Minor comments:

1. Fig. EV3 is not cited in the main text.

Thank you for bringing this to our attention. We have added reference to EV3 in the results section. Note: the panels on this figure now are EV3D and E, due to the addition of new data or panels in EV figures.

2. Limitations are discussed clearly. However, I recommend to discuss other types of on-chip vessels that enable monitoring of long-term vascular damage.

Thank you for the suggestion. Several studies exist which have focused on long-term vascular damage and subsequent recovery. Introduction of a constant or intermittent flow rate in the *in vitro* microvasculature has been proven as a valuable methodology to increase culture time to upwards of a month (Qiu *et al*, 2018; Cherubini *et al*, 2023). Therefore, we stated this and its potential in future iterations of our model as follows:

Line 588: “Previous studies have maintained month-long culture of hydrogel-based *in vitro* microvasculature models by introduction of a constant laminar or intermittent flow, allowing for the study of long-term vascular pathology and recovery (Qiu *et al.*, 2018; Cherubini *et al.*, 2023). Thus, addition of constant or intermittent flow to the model could aid in identification of long-term

disruptive mechanisms involved in CM, such as associated pericyte loss or vascular remodeling.”

4th Sep 2025

Dear Dr. Bernabeu Aznar,

Thank you for the submission of your revised manuscript to EMBO Molecular Medicine. I am pleased to inform you that we will be able to accept your manuscript pending the following final amendments:

- 1) Please address the referee #2 minor point and add the p values as suggested.
- 2) Authors: E-mail correspondence to Mireia Mallorquí could not be delivered. Please update author's e-mail address and make sure to enter correct e-mail addresses for all authors in our submission system.
- 3) Figures:
 - Please remove all figures from the main manuscript file and move their legends at the end of the file. EV figure legends should be placed after main figure legends under the heading "Expanded View Figure Legends".
 - We note that some images/panels are reused. iRBC-egress images from Figure 3A have been reused in Figure 6B. Please cite in the respective figure legend every reused image/panel.
- 4) In the main manuscript file, please do the following:
 - Please address all comments suggested by our data editors listed below:
 - o Data availability statement:
 1. Please note that the specific URL for EMPIAR 11871 dataset is not provided in the data availability statement.
 - o Figure legends:
 1. Please note that the box plots need to be defined in terms of minima, maxima, center, bounds of box and whiskers, and percentile in the legends of figure EV4C.
 2. Please note that information related to n is missing in the legend of figure 4B.
 - Add up to 5 keywords.
 - Please place conflict of interest statement after Acknowledgment under the heading "Disclosure and Competing Interests Statement". We updated our journal's competing interests policy in January 2022 and request authors to consider both actual and perceived competing interests. Please review the policy <https://www.embopress.org/competing-interests> and update your competing interests if necessary.
 - Please remove Reagents and Tools Table and uploaded it as a separate file. Structured Methods section includes Reagents and Tools Table followed by a Methods and Protocols section. More information on how to adhere to this format as well as downloadable templates (.docx) for the Reagents and Tools Table can be found in our author guidelines: <https://www.embopress.org/page/journal/17574684/authorguide#structuredmethods>
An example of a paper with Structured Methods can be found here: <https://www.embopress.org/doi/full/10.1038/s44320-024-00037-6#sec-4>
 - Rename Bibliography to References.
 - Author contributions: Please remove it from the manuscript and specify author contributions in our submission system. CRediT has replaced the traditional author contributions section because it offers a systematic machine-readable author contributions format that allows for more effective research assessment. You are encouraged to use the free text boxes beneath each contributing author's name to add specific details on the author's contribution. More information is available in our guide to authors: <https://www.embopress.org/page/journal/17574684/authorguide#authorshipguidelines>
 - Indicate in legends exact n and exact p values, not a range, along with the statistical test used. To keep the figures "clear" some authors found providing an Appendix table Sx with all exact p-values preferable. You are welcome to do this if you want to.
 - In Data availability statement please use the following format to report the accession number of your data:

[data type]: [full name of the resource] [accession number/identifier] ([doi or URL or identifiers.org/DATABASE:ACCESSION])

Please check "Author Guidelines" for more information.

<https://www.embopress.org/page/journal/17574684/authorguide#availabilityofpublishedmaterial>

5) Synopsis:

- Synopsis image: Please resize the image to 550 px-wide x 300-600 pixels high and upload it as a high-resolution jpeg file. Make sure that text is readable and all the items in the image clearly visible.
 - Please check your synopsis text and image before submission with your revised manuscript. Please be aware that in the proof stage minor corrections only are allowed (e.g., typos).
- 6) As part of the EMBO Publications transparent editorial process initiative (see our Editorial at <http://embomolmed.embopress.org/content/2/9/329>), EMBO Molecular Medicine will publish online a Review Process File (RPF) to accompany accepted manuscripts. This file will be published in conjunction with your paper and will include the anonymous referee reports, your point-by-point response and all pertinent correspondence relating to the manuscript. Let us know whether you agree with the publication of the RPF and as here, if you want to remove or not any figures from it prior to publication. Please note that the Authors checklist will be published at the end of the RPF.
- 7) Please provide a point-by-point letter INCLUDING my comments as well as the reviewer's reports and your detailed

responses (as Word file).

I look forward to reading a new revised version of your manuscript as soon as possible.

Yours sincerely,

Zeljko Durdevic

Zeljko Durdevic
Senior Editor
EMBO Molecular Medicine

*** Instructions to submit your revised manuscript ***

To submit your manuscript, please follow this link:

<https://embomolmed.msubmit.net/cgi-bin/main.plex>

- 1) a .docx formatted version of the manuscript text (including Figure legends and tables)
- 2) Separate figure files*
- 3) supplemental information as Expanded View and/or Appendix. Please carefully check the authors guidelines for formatting Expanded view and Appendix figures and tables at <https://www.embopress.org/page/journal/17574684/authorguide#expandedview>
- 4) a letter INCLUDING the reviewer's reports and your detailed responses to their comments (as Word file).
- 5) The paper explained: EMBO Molecular Medicine articles are accompanied by a summary of the articles to emphasize the major findings in the paper and their medical implications for the non-specialist reader. Please provide a draft summary of your article highlighting
 - the medical issue you are addressing,
 - the results obtained and
 - their clinical impact.This may be edited to ensure that readers understand the significance and context of the research. Please refer to any of our published articles for an example.
- 6) Author contributions: the contribution of every author must be detailed in a separate section.
- 7) EMBO Molecular Medicine now requires a complete author checklist (<https://www.embopress.org/page/journal/17574684/authorguide>) to be submitted with all revised manuscripts. Please use the checklist as guideline for the sort of information we need WITHIN the manuscript. The checklist should only be filled with page numbers where the information can be found. This is particularly important for animal reporting, antibody dilutions (missing) and exact values and n that should be indicated instead of a range.
- 8) Every published paper now includes a 'Synopsis' to further enhance discoverability. Synopses are displayed on the journal webpage and are freely accessible to all readers. They include a short stand first (maximum of 300 characters, including space)

as well as 2-5 one sentence bullet points that summarise the paper. Please write the bullet points to summarise the key NEW findings. They should be designed to be complementary to the abstract - i.e. not repeat the same text. We encourage inclusion of key acronyms and quantitative information (maximum of 30 words / bullet point). Please use the passive voice. Please attach these in a separate file or send them by email, we will incorporate them accordingly.

You are also welcome to suggest a striking image or visual abstract to illustrate your article. If you do please provide a jpeg file 550 px-wide x 300-600px high.

9) A Conflict of Interest statement should be provided in the main text

10) Please note that we now mandate that all corresponding authors list an ORCID digital identifier. This takes <90 seconds to complete. We encourage all authors to supply an ORCID identifier, which will be linked to their name for unambiguous name identification.

Currently, our records indicate that the ORCID for your account is 0000-0001-7212-6209.

Link Not Available

11) Include a Reagents and Tools Table as part of the Methods section, which can be downloaded from our author guidelines (<https://www.embopress.org/page/journal/17574684/authorguide#structuredmethods>)

Photos 400-800 DPI

*Additional important information regarding figures and illustrations can be found at

<https://bit.ly/EMBOPressFigurePreparationGuideline>. See also figure legend preparation guidelines:

<https://www.embopress.org/page/journal/17574684/authorguide#figureformat>

***** Reviewer's comments *****

Referee #2 (Remarks for Author):

My comments have been carefully addressed, and I would like to congratulate the authors on this piece of work.

One minor comment concerns Figure 6C, as the presentation and explanation is somewhat confusing. Does this mean that iRBC-egress with and without rAng-1 is not significantly altered? P-values should be provided for all comparisons in the figure, even (and especially) when not significant.

Referee #3 (Remarks for Author):

The authors addressed all comments clearly. It is ready for publication.

The authors addressed the remaining editorial issues.

29th Sep 2025

Dear Dr. Bernabeu Aznar,

We are pleased to inform you that your manuscript is accepted for publication and is now being sent to our publisher to be included in the next available issue of EMBO Molecular Medicine.

Zeljko Durdevic
Senior Editor
EMBO Molecular Medicine
